# A Benchmark Dataset for Learning from Label Proportions

## Abstract

Learning from label proportions (LLP) has recently emerged as an important technique of weakly supervised learning on aggregated labels. In LLP, a model is trained on groups (a.k.a bags) of feature-vectors and their corresponding label proportions to predict labels for individual feature-vectors. While previous works have developed a variety of techniques for LLP, including novel loss functions, model architectures and their optimization, they typically evaluated their methods on pseudo-synthetically generated LLP training data using common small scale supervised learning datasets by randomly sampling or partitioning their instances into bags. Despite growing interest in this important task there are no large scale open source LLP benchmarks to compare various approaches. Construction of such a benchmark is hurdled by two challenges a) lack of natural large scale LLP like data, b) large number of mostly artificial methods of forming bags from instance level datasets.

In this paper we propose *LLP-Bench*: a large scale LLP benchmark constructed from the Criteo Kaggle CTR dataset. We do an in-depth, systematic study of the Criteo dataset and propose a methodology to create a benchmark as a collection of diverse and large scale LLP datasets. We choose the Criteo dataset since it admits multiple natural collections of bags formed by grouping subsets of its 26 categorical features. We analyze all bag collections obtained through grouping by one or two categorical features, in terms of their bag-level statistics as well as embedding based distance metrics quantifying the geometric separation of bags. We then propose to include in LLP-Bench a few groupings to fairly represent real world bag distributions.

We also measure the performance of state of the art models, loss functions (adapted to LLP) and optimizers on LLP-Bench. We perform a series of ablations and explain the performance of various techniques on LLP-Bench. To the best of our knowledge LLP-Bench is the first open source benchmark for the LLP task. We hope that the proposed benchmark and the evaluation methodology will be used by ML researchers and practitioners to better understand and hence devise state of art LLP algorithms.

## 1 Introduction

In traditional supervised learning, *training* data consists of feature-vectors (instances) along with their labels. A model trained using such data is then used during inference to predict the labels of *test* instances. In recent times, primarily due to privacy concerns and relative rarity of high quality supervised data, the weakly supervised framework of *learning from label proportions* (LLP) has gained importance (Scott & Zhang (2020); Saket et al. (2022); O'Brien et al. (2022)). In LLP, the training data is available as a collection of subsets or *bags* of instances along with the *label proportion* for each bag. The goal is to learn a classification model for predicting the class-labels of individual instances (de Freitas & Kück (2005); Musicant et al. (2007)).

Clearly, supervised learning is the special case of LLP when all bags are unit-sized. Unlike supervised learning however, for which a multitude of task-specific real-world datasets are easily available, the same is not true for LLP. While previous works have developed and explored a variety of algorithmic, optimization and deep-neural net based techniques for LLP (see Sec. 2 for more de-

tails), all of them experimentally evaluate their methods on *pseudo-synthetic* LLP datasets consisting instances of some supervised learning dataset randomly sampled/partitioned into the different bags. Further, most of the above works use limited scale data, typically small UCI (Dua & Graff (2017)), image and social media datasets.

An exception to the above is the work of Saket et al. (2022) which also uses the Criteo Kaggle CTR (Criteo (2014)) and MovieLens-20m (Movielens-20M; Harper & Konstan (2016)) which are fairly large in scale, roughly 45 million instances and 20 million instances respectively. In particular, the Criteo dataset has 13 numerical and 26 categorical features whose semantics are undisclosed. Each row is an impression and a $\{0, 1\}$-valued label indicates a click, with in total 7 days of impression-click data. The categorical features can be used to create many different bag collections depending on their subset used for grouping, where each choice of the subset's values yields a bag of instances having those feature values. These groupings simulate the typical aggregation scenarios in real-world use-cases, however Saket et al. (2022) only experimented in a limited manner with one grouping.

In contrast to the above state of affairs, a large number of publicly accessible real-world and large scale supervised-learning datasets have been studied over the years, whereas there are hardly any datasets which are curated specifically for LLP.

## 1.1 OUR CONTRIBUTIONS

In this work we address the unavailability of a large scale benchmark and standardized evaluation methodologies for LLP. We make the following contributions in this paper towards creating an LLP benchmark building on top of the publicly available Criteo Kaggle CTR dataset.

**Bag collections using group-by feature-sets.** Typically, for privacy preservation in CTR applications, the impressions are grouped into bags according the values of features such as advertiser-id, product-id, date etc. Thus, we can simulate such aggregations on the Criteo dataset using any subset of the categorical feature-set. However, we observe that choosing more than three categorical features likely results in small-sized bags which would be contrary to the goal of large scale LLP datasets. Therefore, our exploration limits the groupings to those obtained using at most two of the categorical features. Below we present the different aspects of our exploration of these groupings. We use a standard preprocessing previously used for training the AutoInt model (Song et al. (2019)) on the Criteo dataset at the instance level. More details can be found in Section 4.

**Analysis, categorization, and filtering groupings.** There are 26 categorical features, leading to $26 + \binom{26}{2} = 351$ possible groupings using at most two categorical features. Our goal is to curate LLP datasets with not too small or very large bags (as the latter have very weak label supervision), and we always remove bags of size $\leq 50$ and those of size $> 2500$ from these groupings, similar to the work of Saket et al. (2022). Post this removal, we identify as outliers those groupings which have at most 500 bags. The remainder 308 groupings are further analyzed in relation to their bag size and label proportion distributions. For each grouping We calculate the threshold bags sizes such that $t\%$ of the bags have at most that size, for $t = 50, 70, 85, 95$. Using normalized vectors of these four values, we use $k$-Means clustering to the partition the groupings into four subsets typified by increasing bag sizes. More details of these clusters can be found in Sec 5.1. Further, modeling the labels as iid Bernoulli with bias given by the average label of the dataset, we compute for each grouping the average of the log likelihoods of the bag label proportions. Using this we also cluster the set of groupings into four subsets indicating how far-from random their label proportions are. Analysis of this characterisation can be found in Section 5.2.

In the above removal of bags, a substantial fraction of the original dataset is also removed since there is an abundance of small bags for most groupings. For subsequent analysis involving model training, we further filter out those ones which retain less than 30% of the instances. This is to ensure that we only have large-scale LLP bag collections, and we obtain 52 groupings satisfying the retention condition. Details of removals by these filters can be found in Section 4.2.

It turns out that these groupings have a similar number and average size of bags. We then proceed to estimate the geometric clustering of bags by computing the average *inter-bag* and *intra-bag* distances for these groupings. For this we use natural definitions of these notions based on the

$\ell_2^2$-distance for ease of computation. We also prove certain metric like properties for the inter-bag distances. Details and analysis of these distances is present in Section 5.3.

From the above analyzed groupings we select a representative subset with diverse statistical properties and include them in *LLP-Bench* - our proposed LLP benchmark as a collection of LLP datasets.

**LLP model training methodology** For each of the 52 grouping we create a LLP model training and testing setup as follows. We remove the small and large bags as above, and then recreate the instance-level dataset out of the remaining bags. We apply 5-fold split to obtain 5 pairs of train/test splits at the instance-level. For each train/test split, the training bags are recreated using the same grouping on the train set. We then train a 1-Layer MLP, 2-layer MLP and the AutoInt models using various hyperparameter and optimizer settings, on the training bags and evaluate w.r.t AUC scores on the test set. Details of the training and reported AUC scores are present in Section 6.1.

**Statistically correlating LLP model performance.** From the above experimentation we obtain statistics for different groupings based on their bag and label proportion distributions. We calculate the Pearson correlation scores with the model training performance of the LLP data-statistics for the different groupings. The main observations are:

1. A negative correlation with the percentile bag size thresholds indicating that the model performance degrades when the groupings have larger bags. This is intuitively consistent with larger bags having less label supervision (for the same label proportion) that smaller bags.

2. A negative correlation with the label proportion log likelihood. Since bags with label proportions deviating from the global label bias have lower likelihood, roughly speaking this means that those groupings where the positive labels are concentrated in fewer bags have better training performance.

3. A positive correlation with the ratio of Average Inter-Bag Distances and Average Intra-Bag Distances. Higher ratio indicates a good separation between the bags. Hence, positive correlation indicates that models perform better when there is considerable variation in bag distributions.

Some other correlation scores obtained and their interpretation is described in Section 6.2.

## 2   PREVIOUS WORK

The study of LLP is motivated by applications in which only the aggregated labels for groups (bags) of feature vectors are available due to privacy or legal (Rüping (2010),Wojtusiak et al. (2011)) constraints or inadequate or costly supervision Dery et al. (2017); Chen et al. (2004). It has also been used for several weakly supervised tasks such as IVF prediction (Hernández-González et al. (2018)) and image classification (Bortsova et al. (2018); Ørting et al. (2016)). More recently, LLP has been proposed by O'Brien et al. (2022) as a framework for privacy preserving conversion prediction.

Several techniques for LLP have been studied over the years. de Freitas & Kück (2005); Hernández-González et al. (2013) applied trained probabilistic models using Monte-Carlo methods. Subsequent works (Musicant et al. (2007); Rüping (2010)) extended supervised learning techniques such neural nets, SVM and *k-nearest neighbors* to LLP, others adapted clustering based approaches (Chen et al. (2009); Stolpe & Morik (2011)), while Yu et al. (2013) proposed a novel $\propto$-SVM method for LLP. Quadrianto et al. (2009) estimated model parameters from label proportions for the exponential generative model and assumptions on label distributions of bags. Their method was further applied by Patrini et al. (2014) for more general models and relaxed distributional assumptions. More recent works have investigated deep neural network based LLP methods (Bortsova et al. (2018); Ardehaly & Culotta (2017); Liu et al. (2019)) and techniques using bag combinations (Scott & Zhang (2020); Saket et al. (2022)). Recently, Saket (2021) initiated a theoretical study of LLP from the computational learning perspective.

All of the previous works in LLP experimentally evaluate their methods on LLP datasets consisting of bags randomly created from some real-world supervised learning dataset. In these *pseudo-synthetic* LLP datasets, instances are randomly sampled/partitioned into the different bags, where in Patrini et al. (2014) and Saket et al. (2022) this process also clusters feature-vectors to generate more

complicated bag distributions. Further, most of the above works use limited scale data, typically small to medium scale UCI datasets (Yu et al. (2013); Patrini et al. (2014); Scott & Zhang (2020)), image datasets (Liu et al. (2019)), social media data (Ardehaly & Culotta (2017)) etc. In general, the have been very few *natural*, real-world LLP datasets used for evaluations in previous works. As mentioned earlier Saket et al. (2022) experimented with MovieLens-20m and Criteo datasets. They used a temporal aggregation into bags for MovieLens-20m, while on the Criteo dataset they only used one pair of categorical features to create a collection of bags for experimentation.

## 3 PRELIMINARIES

In our exploration of LLP we shall only consider binary $\{0, 1\}$-valued instance labels.

**Notation**: $X := \{\mathbf{x}^{(i)} \in \mathbb{R}^n\}_{i=1}^m$ is the dataset of $m$ feature vectors in $n$-dimensional space with labels given by $Y := \{y^{(i)} \in \{0, 1\}\}_{i=1}^m$. We denote by $\hat{Y} := \{\hat{y}^{(i)} \in [0, 1]\}_{i=1}^m$ the corresponding model predictions which are probabilities of the predicted label being 1.

**Definition 3.1** (Bag). *A bag $B \subseteq [m]$ consists of feature vectors $X_B := \cup_{i \in B}\mathbf{x}^{(i)}$ and with the corresponding label histogram $y_B := \Sigma_{i \in B}y^{(i)}$. The label proportion of the bag is $y_B/|B|$.*

**Definition 3.2** (LLP Dataset). *A learning from label proportions (LLP) dataset corresponding to a collection of bags $\mathcal{B} := \{B_j\}_{j=1}^{j=N}$ is given by $\{(X_B, y_B) \mid B \in \mathcal{B}\}$. The label bias of training dataset is $\mu(\mathcal{B}, Y) := \left(\sum_{B \in \mathcal{B}} y_B\right) / \left(\sum_{B \in \mathcal{B}} |B|\right)$.*

Since the LLP training dataset lacks instance-level labels we use the dataset label bias to model the label histogram of a bag $B$ as the binomial distribution $\mathrm{B}(|B|, p)$ where $p = \mu(\mathcal{B}, Y)$. Its *log-likelihood* is $\mathrm{LL}(B, y_B) = \log f(|B|, y_B, p)$ where $f(r, k, p) := \binom{r}{k} p^k (1 - p)^{r-k}$ is the pdf of $\mathrm{B}(r, p)$. The dataset average bag log-likelihood is $\mathrm{AvgBagLL}(\mathcal{B}, Y) := \left(\sum_{B \in \mathcal{B}} \mathrm{LL}(B, y_B)\right) / |\mathcal{B}|$.

### 3.1 LLP MODEL TRAINING

In order to train a model on an LLP dataset, we apply common loss functions at the bag level. In this work we experiment with the *binary cross entropy* loss $\mathsf{L}_{\mathrm{bce}}$ and the *mean-squared error* loss $\mathsf{L}_{\mathrm{mse}}$, which are define for a bag $B$ with with average label $z_B := y_B/|B|$ and average label prediction $\hat{z}_B := \hat{y}_B/|B|$ as:

$$\mathsf{L}_{\mathrm{bce}}(B, z_B, \hat{z}_B) := -\left(z_B \log \hat{z}_B + (1 - z_B) \log(1 - \hat{z}_B)\right) \ , \ \mathsf{L}_{\mathrm{mse}}(B, y_B, \hat{y}_B) := |y_B - \hat{y}_B|^2. \quad (1)$$

Note that both the above losses are minimized when $\hat{y}_B = y_B$.

In our experiments we use mini-batch based model training on LLP dataset. A mini-batch here consists of $k$ bags $B_1, \ldots, B_k$ and their corresponding label histograms $y_{B_1}, \ldots, y_{B_k}$. The model predicts on all the instances in the bags of the minibatch are aggregated into the predicted label histograms $\hat{y}_{B_1}, \ldots \hat{y}_{B_k}$. The batch-level loss is given by sum of the bag-level losses over the for the mini-batch bags.

### 3.2 BAG-LEVEL DISTANCES

We also analyse the geometric clustering of the feature vectors in bags, by comparing the separation among feature-vectors within bags and their separation across bags. For this, we define a natural bag separation.

**Definition 3.3** (Bag Separation). *For a distance $d$ on $\mathbb{R}^n$ and collection of bags $\mathcal{B} = \{B_1, \ldots, B_M\}$ the corresponding separation function is defined as $\mathsf{BagSep}(B, B', d) := \frac{1}{|B||B'|}\Sigma_{\mathbf{x} \in B}\Sigma_{\mathbf{x}' \in B'}d(\mathbf{x}, \mathbf{x}')$. We define the $M \times M$ matrix $\mathsf{BagSepMatrix}(\mathcal{B}, d)$ whose $(i, j)$th element is given by $\mathsf{BagSep}(B_i, B_j, d)$.*

While $\mathsf{BagSep}$ is not a metric since $\mathsf{BagSep}(B, B)$ is not necessarily zero, the following lemma (proved in Appendix A.1) shows that it does satisfy the other metric properties.

**Lemma 3.4.** $\mathsf{BagSep}$ *satisfies non-negativity, symmetry and triangle inequality.*

We use BagSep to compute the average separation between pairs of bags and the average separation within each bag. If the feature-vectors in bags are clustered together and far away from those of other bags, we expect the former to be significantly greater than the later.

**Definition 3.5** (Inter-Bag Separation for a bag). *Given $\mathcal{B}$, and metric $d$ on $\mathbb{R}^n$, the average inter-bag distance for a bag $B \in \mathcal{B}$ is defined as* $\mathsf{InterBagSep}(B, d) := \frac{1}{|\mathcal{B}|-1} \sum_{B' \in \mathcal{B}, B' \neq B} \mathsf{BagSep}(B, B', d)$.

For computing the average statistic for the entire dataset we define the following.

**Definition 3.6.** *The mean intra-bag separation of $\mathcal{B}$ is defined as* $\mathsf{MeanIntraBagSep}(\mathcal{B}, d) := \frac{1}{|\mathcal{B}|} \sum_{B \in \mathcal{B}} \mathsf{BagSep}(B, B, d)$. *The mean of average inter-bag separation is* $\mathsf{MeanInterBagSep}(\mathcal{B}, d) := \frac{1}{|\mathcal{B}|} \sum_{B \in \mathcal{B}} \mathsf{InterBagSep}(B, d)$.

We have the following lemma proved in Appendix A.2.

**Lemma 3.7.** *For any bag $B$, (i)* $\mathsf{InterBagSep}(B, d)/\mathsf{BagSep}(B, B, d) \geq 1/2$ *when $d$ is a metric, (ii)* $\mathsf{InterBagSep}(B, d)/\mathsf{BagSep}(B, B, d) \geq 1/4$ *when $d$ is the $\ell_2^2$ distance.*

The following is a straightforward corollary of Lemma 3.7.

**Corollary 3.8.** *(i)* $\mathsf{MeanInterBagSep}(\mathcal{B}, d)/\mathsf{MeanIntraBagSep}(\mathcal{B}, d) \geq 1/2$ *when $d$ is a metric, (ii)* $\mathsf{MeanInterBagSep}(\mathcal{B}, d)/\mathsf{MeanIntraBagSep}(\mathcal{B}, d) \geq 1/4$ *when $d$ is the $\ell_2^2$ distance.*

We expect this ratio to achieve values substantially less than 1 in adversarial cases. Appendix A.5 provides an example of such a case. For convenience, for $\mathcal{B}$, we use InterIntraRatio to denote $\mathsf{MeanInterBagSep}(\mathcal{B}, d)/\mathsf{MeanIntraBagSep}(\mathcal{B}, d)$ when $d = \ell_2^2$.

## 3.3 CRITEO DATASET

The Criteo CTR dataset (Criteo (2014)) has 13 numerical and 26 categorical features and a binary label. Each of the approximately 45 million rows (instances) represents an impression (online ad) and the label indicates a click. The semantics of all the features is undisclosed and the values of all the categorical features hashed into 32-bits for anonymization. Additionally, the dataset has missing values. We use a preprocessed version of the dataset as done for the AutoInt (Song et al. (2019)) model, described and implemented in their provided code[1]. For convenience we label the numerical and categorical features (in their order of occurrence) as $N1, \ldots, N13$ and $C1, \ldots C26$. The preprocessing applies $\mathrm{int}(\log^2(x))$ transformation when $x > 2$ on the numerical feature values $x$, and we further additively scale so that their values are non-negative integers. The categorical features are encoded as non-negative integers.

## 4 LLP DATASET: BAG CREATION

We create the LLP dataset by grouping the instances by subsets $\mathcal{C} \subseteq \{C1, \ldots, C26\}$ of the categorical columns, where $\mathcal{C} \leq 2$. For each setting of the values of $\mathcal{C}$ we obtain a bag with instances with those values of $\mathcal{C}$. Each such grouping yields an LLP dataset[2]. Thus, we obtain $\binom{26}{2} + 26 = 351$ LLP datasets, each referred to also as a *grouping* on $\mathcal{C}$ ($|\mathcal{C}| \leq 2$). Note that for any grouping, the set of bags partition the dataset and therefore each instance occurs in exactly one bag.

### 4.1 CLIPPING GROUPINGS FOR BAG DISTRIBUTION ANALYSIS

As mentioned in Sec. 1.1 we *clip* the groupings by discarding all bags of sizes less than 50 and greater than 2500, as our goal is to analyze reasonable LLP datasets. We observe that some groupings are left with with very few bags or zero bags, while others have a large number of bags. For e.g., the initial grouping on $C9$ creates only 3 bags and the grouping on $C20$ creates only 4 bags. Hence, after clipping thse groupings have no bags. The groupings on $C6$, $C17$, $C22$, $C23$ and $\{C9, C20\}$

---

[1]The url is `https://github.com/DeepGraphLearning/RecommenderSystems/tree/master/featureRec`.

[2]Note that for model training purposes such bags may be created from only the *train set* portion of the entire dataset

all contain less than 20 bags. On the other hand, groupings on $\{C10, C16\}$ and $\{C4, C10\}$ each contain more than $8 \times 10^6$ bags.

We compute the mean bag sizes of the clipped groupings. The lowest mean bag size is 62 which we obtain is for the clipped grouping on $C23$. It manages to retain just one bag after clipping and has 62 instances in that bag. Similarly, the highest mean bag size that we obtain is 1292 obtained on clipped grouping on $C17$. It also retains a single bag after clipping with 1292 instances in that bag. Table 4 provides these statistics for a sample of the groupings. Refer Appendix A.8 for statistics of all groupings.

The bag distribution analysis described in Sec. 5 is performed on the 308 clipped groupings with at least 500 bags remaining.

### 4.2 FILTERING GROUPINGS FOR MODEL TRAINING

We apply the following filters on the clipped groupings to choose groupings for model training.

*Label Information Loss Filter.* If the number of bags that remain is less that 10000, we discard such groupings to ensure sufficient number training bags. After applying this filter, we are left with 240 groupings.

*Instance Information Loss Filter.* We drop a grouping if it is left with less than 30% of the original number of instances ($\approx 13.75 \times 10^6$ instances). After applying this filter, we are left with 52 groupings. All the groupings in single columns are filtered out as the maximum percentage of instances any of these groupings retains is 21.68% ($C4$). We finally obtain a set of 52 groupings which satisfy both of the conditions listed above, all of which are emboldened in Table 10 in the Appendix.

## 5 BAG DISTRIBUTION ANALYSIS

We perform the bag distribution analysis for all 308 clipped groupings which contain more than 500 bags.

### 5.1 CHARACTERISING THE DISTRIBUTION OF BAG SIZES

Since we have only have a label proportion for each bag, informally speaking, the larger the bag size the lower the amount of label supervision for that bag. The bag sizes for any grouping are characterized by their cumulative distribution function which plots the fraction of bags of size at most $t$ for all $t \geq 1$. In all the groupings, it is observed that the density of bags drops steeply with the increase in bag size in the histograms of bag sizes. Thus, we compute the bag sizes at the 50, 70, 85 and 95 percentile of cumulative distribution plot, for each grouping.

Hence, we can naturally classify the groupings we obtain into *long-tailed* and *short-tailed* distributions. Short-tailed distributions have most bags of small size and a very few large sized bags whereas Long-tailed distributions contain many bags of large sizes. Bags of large sizes provide a very little label information for a lot of feature level information. Hence, they can be used for learning representations but are less useful in supervised training.
In order to classify the groupings created into *long-tailed* and *short-tailed*, we compute the threshold bag sizes at which we attain 50, 70, 85 and 95 percentile of the bags for each clipped grouping. We normalize these values and obtain 4-dimensional vectors for each clipped grouping. Applying $k$-Means on these vectors we cluster the clipped groupings into 4 clusters. As shown in Table 1, the mean $t$-percentile bag size, give the same cluster ordering for $t = 50, 70, 85, 95$. Hence, we name the clusters in increasing order of these mean bag sizes as *Very Short-tailed*, *Short-tailed*, *Long-tailed* and *Very Long-tailed* bag size distributions. Appendix A.9 contains threshold bag sizes and cluster labels based on them for all groupings.

### 5.2 CHARACTERISING THE DISTRIBUTION OF LABEL HISTOGRAMS

We model the distribution of label histograms in a grouping as a binomial distribution, with bias as the label proportion of the grouping. We compute for each grouping its AvgBagLL value. The

Table 1: Mean bag sizes at which groupings achieve 50, 70, 85 and 95 percentile in each cluster

| Bag Size Dist. Cluster | # Groupings | Mean bag size: 50 Percentile | Mean bag size: 70 Percentile | Mean bag size: 85 Percentile | Mean bag size: 95 Percentile |
|---|---|---|---|---|---|
| *Very Short-tailed* | 171 | 107.77 | 189.05 | 375.94 | 905.66 |
| *Short-tailed* | 82 | 155.40 | 314.83 | 672.02 | 1434.77 |
| *Long-tailed* | 41 | 269.22 | 571.24 | 1094.90 | 1831.78 |
| *Very Long-tailed* | 14 | 590.79 | 1053 | 1599.57 | 2145.93 |

Table 2: Clustering on AvgBagLL.

| Cluster | # Groupings | Min LL | Max LL |
|---|---|---|---|
| *High* | 26 | -9.48 | -3.26 |
| *Medium* | 73 | -15.98 | -9.8 |
| *Low* | 173 | -24.19 | -16.02 |
| *Very Low* | 36 | -38.88 | -25.47 |

Table 3: Clustering on InterIntraRatio.

| Cluster | # Groupings | Min Ratio | Max Ratio |
|---|---|---|---|
| *Less-separated* | 37 | 1.02 | 1.10 |
| *Medium-separated* | 196 | 1.12 | 1.24 |
| *Well-separated* | 58 | 1.25 | 1.40 |
| *Far-separated* | 17 | 1.41 | 1.56 |

higher the value, the closer the grouping is to having randomly distributed label proportions. Refer to Sec. 3 for the definitions of AvgBagLL and bias of the dataset.

We perform $k$-Means on these values and classify our groupings into groupings having *High*, *Medium*, *Low* and *Very Low* AvgBagLL. The ranges of AvgBagLL values for each of these clusters are listed in Table 2. Appendix A.10 contains the AvgBagLL values as well as the clusters labels based on them for all groupings.

### 5.3 BAG SEPARATION ANALYSIS

As defined in Sec. 3, higher InterIntraRatio indicates bags are clustered. First, we obtain an embedding of the feature-vectors by training an instance level AutoInt Model[3] on *Criteo* dataset and extract it's embedding layer using which we transform the instances into this embedding space. We compute the InterIntraRatio and other BagSep quantities for all 308 clipped groupings in this embedding space. We then perform $k$-Means on InterIntraRatio to classify our groupings into *Less-separated*, *Medium-separated*, *Well-separated* and *Far-separated* bag distributions.

The Ranges of InterIntraRatio for each of the clusters are listed in Table 3. Appendix A.11 contains the BagSep quantities for all the groupings along with the clusters they are classified into. Appendix A.4 contains simplification of BagSep computation with $\ell_2^2$-distance.

### 5.4 LLP-BENCH: A REPRESENTATIVE COLLECTION OF LLP DATASETS

Table 4 provides a representative set of groupings, with their cluster assignments (as per the various analyses above) along with bag-level statistics. Several of these groupings are also used for the LLP model training and analysis presented below. We propose LLP-Bench as collection of LLP datasets with naturally constructed bags which simulate real-world LLP use-cases and can be used for evaluating LLP techniques.

## 6 MODEL PERFORMANCE ON TRAINABLE GROUPINGS

### 6.1 TRAINING METHODOLOGY

We train on 52 clipped groupings which pass the filters in Sec. 4.2, for which we create the train and test sets as follows. For each grouping, we recreate the instance-level dataset from the clipped bag-level dataset and the original labels. On this truncated instance-level dataset we perform a 5-Fold split, and for each split we obtain the training bags dataset by grouping the train set on the same categorical features. The test sets remain at the instance-level.

We train 1-Layer MLP, 2-Layer MLP[4] and the AutoInt model. Preprocessing mentioned in 1.1 ensures that all features have non-negative integers. We use a multihot layer whose output is passed

---

[3]80% instances used for training and rest for validation.

[4]1-Layer MLP has 64 hidden units, 2-Layer MLP has 128 and 64 units in successive layers, tanh activation

Table 4: LLP-Bench Groupings. Bold : Analyzed for model training.

| Col1 | Col2 | No. Bags After Clipping | Percentage Inst. After Clipping | Mean Bag Size | Bag Distribution Clusters | Label Prop Dist. Clsuters | Inter/Intra Ratio Clusters |
|---|---|---|---|---|---|---|---|
| *C5* | *C8* | 2486 | 1.44 | 265.84 | *Very Short-tailed* | *High* | *Less-separated* |
| ***C1*** | ***C10*** | **55528** | **32.44** | **267.81** | ***Very Short-tailed*** | ***High*** | ***Medium-separated*** |
| *C19* | *C22* | 3893 | 3.69 | 434.23 | *Long-tailed* | *High* | *Less-separated* |
| ***C6*** | ***C10*** | **46981** | **31.8** | **310.24** | ***Short-tailed*** | ***Medium*** | ***Medium-separated*** |
| ***C2*** | ***C13*** | **45206** | **34.87** | **353.59** | ***Short-tailed*** | ***Low*** | ***Medium-Separated*** |
| ***C7*** | ***C10*** | **56575** | **40.52** | **328.31** | ***Short-tailed*** | ***Medium*** | ***Far-separated*** |
| ***C10*** | ***C15*** | **102841** | **44.66** | **199.07** | ***Very Short-tailed*** | ***Medium*** | ***Well-separated*** |
| ***C7*** | ***C21*** | **88970** | **40.43** | **208.31** | ***Very Short-tailed*** | ***Low*** | ***Far-separated*** |
| *C13* | - | 1221 | 2.98 | 1120.01 | *Very Long-tailed* | *Very Low* | *Less-separated* |
| *C9* | *C19* | 2214 | 2.6 | 538.05 | *Long-tailed* | *Medium* | *Far-separated* |
| ***C7*** | ***C10*** | **56575** | **40.52** | **328.31** | ***Short-tailed*** | ***Medium*** | ***Far-separated*** |
| *C9* | *C11* | 3221 | 5.71 | 812.21 | *Very Long-tailed* | *Very Low* | *Far-separated* |
| ***C7*** | ***C20*** | **30420** | **31.85** | **479.96** | ***Long-tailed*** | ***Low*** | ***Well-separated*** |

Table 5: AUC Scores of MLP classifiers obtained on LLP-Bench Groupings

| Col1 | Col2 | E1 | E2 | E3 | E4 | E5 | E6 | E7 | E8 |
|---|---|---|---|---|---|---|---|---|---|
| *C1* | *C10* | 73.64±0.05 | 73.67±0.05 | 73.6±0.06 | 73.61±0.04 | 65.03±0.11 | 63.43±0.1 | 65.55±0.22 | 63.97±0.23 |
| *C6* | *C10* | 72.89±0.04 | 72.9±0.05 | 72.79±0.04 | 72.81±0.05 | 65.01±0.08 | 62.88±0.31 | 65.49±0.16 | 63.84±0.44 |
| *C2* | *C13* | 75.23±0.04 | 75.23±0.03 | 75.06±0.03 | 75.08±0.04 | 68.04±0.07 | 66.75±0.13 | 68.57±0.21 | 67.21±0.12 |
| *C7* | *C10* | 73.49±0.03 | 73.5±0.02 | 73.35±0.03 | 73.31±0.03 | 64.32±0.16 | 62.66±0.17 | 65.23±0.27 | 63.2±0.19 |
| *C10* | *C15* | 75.76±0.04 | 75.75±0.04 | 75.58±0.05 | 75.56±0.04 | 70.07±0.05 | 68.83±0.11 | 70.63±0.08 | 69.51±0.17 |
| *C7* | *C21* | 76.89±0.04 | 76.91±0.04 | 76.64±0.04 | 76.67±0.06 | 70.87±0.08 | 69.76±0.18 | 71.48±0.05 | 70.38±0.13 |
| *C7* | *C10* | 73.49±0.03 | 73.5±0.02 | 73.35±0.03 | 73.31±0.03 | 64.32±0.16 | 62.66±0.17 | 65.23±0.27 | 63.2±0.19 |
| *C7* | *C20* | 74.43±0.06 | 74.46±0.06 | 74.28±0.05 | 74.28±0.08 | 64.65±0.12 | 63.09±0.2 | 65.21±0.2 | 63.64±0.1 |

to the MLP models. AutoInt has a 16-dimensional trainable embedding layer for each feature[5]. Output layer has one unit with sigmoid activation.

We perform minibatch gradient descent by sampling minibatches of 8 bags each, and the model predictions are aggregated into the predicted label proportions of the bags. The minibatch loss is the sum over the bags of either the bag-level $L_{mse}$ or $L_{bce}$ as described in Sec. 3.1. We then back-propagate this loss and update weights using the optimizer – either Adam or SGD. The specifications of experiments are in Table 7. Using Adam, we train for $50$ epochs with a learning rate of $1e-5$ for initial 15 epochs and $1e-6$ for the rest. Using SGD, we train for 300 epochs with constant learning rate of $1e-5$.

We use test AUC scores to qualify the tractability of an LLP dataset. For MLP trained using SGD and Adam we use the maximum reported AUC score. On the other hand, AutoInt has an increasing trend for both optimizers but it is not (even locally) monotonic, hence for it we use the average of last 5 epochs of training. The AUC score (averaged over the 5 splits) for trainable groupings in *LLP-Bench* and the various experiments (see in conjunction with Table 7) are listed in 5 and Table 6. AUC score for all trainable groupings are listed in Table 8. Appendix A.7 contains details of instance-level training which we perform for completeness.

Table 6: AUC Scores of AutoInt obtained on LLP-Bench Groupings

| Col1 | Col2 | E9 | E10 | E11 |
|---|---|---|---|---|
| *C1* | *C10* | 70.4±0.17 | 70.23±0.18 | 61.98±0.38 |
| *C6* | *C10* | 69.84±0.24 | 69.75±0.22 | 63.3±1.28 |
| *C2* | *C13* | 68.81±0.23 | 68.88±0.42 | 64.51±1.74 |
| *C7* | *C10* | 69.11±0.34 | 68.77±0.28 | 62.31±2.02 |
| *C10* | *C15* | 71.88±0.11 | 71.93±0.02 | 69.52±0.19 |
| *C7* | *C21* | 72.5±0.11 | 72.84±0.16 | 71.49±1.01 |
| *C7* | *C10* | 69.11±0.34 | 68.77±0.28 | 62.31±2.02 |
| *C7* | *C20* | 67.98±0.82 | 67.79±0.24 | 63.06±0.89 |

Table 7: Experiment Legend

| Expt. | Model | Opt. | Loss |
|---|---|---|---|
| *E1* | 1-Layer MLP | Adam | BCE |
| *E2* | 1-Layer MLP | Adam | MSE |
| *E3* | 2-Layer MLP | Adam | BCE |
| *E4* | 2-Layer MLP | Adam | MSE |
| *E5* | 1-Layer MLP | SGD | BCE |
| *E6* | 1-Layer MLP | SGD | MSE |
| *E7* | 2-Layer MLP | SGD | BCE |
| *E8* | 2-Layer MLP | SGD | MSE |
| *E9* | AutoInt | Adam | BCE |
| *E10* | AutoInt | Adam | MSE |
| *E11* | AutoInt | SGD | BCE |

---

[5]Embedding Layer for categorical features, numerical features multiplied by trainable 16-dim vector

## 6.2 CORRELATION OF DATASET CHARACTERISTICS WITH AUC SCORES

We compute the Pearson correlation between the AUC scores and the bag level statistics computed in Sec. 5. These are visualised in Fig. 1. Some observations from these scores are:

1. Positive correlation with number of bags and number of instances. This is as expected as each bag adds to the label information and each instance adds to feature information available to the classifier.

2. Negative correlation with the mean bag size and percentile bag size thresholds. This is intuitively consistent with larger bags having less label supervision (for the same label proportion) than smaller bags, and typically the model performance would degrade when the groupings have larger bags.

3. Negative correlation with the label proportion log likelihood. A lower log likelihood indicates that labels proportions in the dataset are highly skewed. This means that those groupings where the positive labels are concentrated in fewer bags have better training performance. In this case, the bag grouping features provide significant supervision which the model can leverage. We can infer the same from highly positive correlation with standard deviation of label proportion.

4. Positive correlation with the InterIntraRatio. Higher ratio indicates a good separation between the bags in input space. Hence, positive correlation indicates that models perform better when bags are separable in input space. This can be explained as follows

   – The distribution of label proportions are skewed as the maximum log-likelihood exhibited is $-3.26$. Hence, substantial label information is present at the bag-level.
   
   – If the InterIntraRatio is high, much of the discriminative information at the bag-level lies in the input space itself. If the InterIntraRatio is low, most of this information is in some latent space that the model needs to learn.

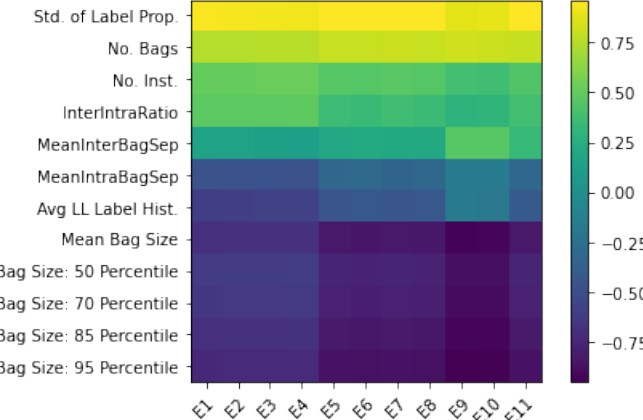

Figure 1: Correlation Heatmap

## 7 CONCLUSION

Our work conducts an in-depth study of the Criteo CTR dataset for use as a *natural* LLP dataset, and provides LLP-Bench: a collection of LLP datasets from from the Criteo dataset as a benchmark for evaluating LLP techniques. In this process, our work analyzes bag collections given by grouping on at most two categorical features, based on their distribution of bags as well as label proportions. We also adopt an evaluation methodology using which we train various models on an appropriately filtered subset of groupings and demonstrate (as well explain) correlation of the model performance with the computed statistics.

We believe our work addresses to a great extent the current lack of natural LLP benchmarks, and provides LLP-Bench using which LLP techniques can be systematically evaluated.

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

# Appendix for: A Benchmark Dataset for Learning from Label Proportions

## A    PROOFS OF LEMMAS AND ALGORITHMS

### A.1    PROOF OF LEMMA 2.4

*Proof.* From Def. 3.3, the non-negativity and symmetry properties are obvious.

**Triangle Inequality** : let $B_1, B_2, B_3 \in \mathcal{B}$, and we use the following notation for convenience: $B_1 = \{x_i | i \in [n]\}$, $B_2 = \{y_j | j \in [m]\}$, $B_3 = \{z_k | k \in [l]\}$. As $d$ is a metric, we know that for all $i \in [n], j \in [m]$ and $k \in [l]$, $d(x_i, z_k) \le d(x_i, y_j) + d(y_j, z_k)$. Hence,

$$d(x_i, z_k) \le \frac{\Sigma_{j=1}^{j=m} d(x_i, y_j)}{m} + \frac{\Sigma_{j=1}^{j=m} d(y_j, z_k)}{m}$$

$$\Rightarrow \quad \frac{\Sigma_{i=1}^{i=n} d(x_i, z_k)}{n} \le \frac{\Sigma_{i=1}^{i=n} \Sigma_{j=1}^{j=m} d(x_i, y_j)}{nm} + \frac{\Sigma_{j=1}^{j=m} d(y_j, z_k)}{m}$$

$$\Rightarrow \quad \frac{\Sigma_{k=1}^{k=l} \Sigma_{i=1}^{i=n} d(x_i, z_k)}{ln} \le \frac{\Sigma_{i=1}^{i=n} \Sigma_{j=1}^{j=m} d(x_i, y_j)}{nm} + \frac{\Sigma_{k=1}^{k=l} \Sigma_{j=1}^{j=m} d(y_j, z_k)}{ml}$$

$$\Rightarrow \quad \mathsf{BagSep}(B_1, B_3, d) \le \mathsf{BagSep}(B_1, B_2, d) + \mathsf{BagSep}(B_2, B_3, d)$$

$\square$

## A.2 Proof of Lemma 2.7

*Proof.* Let $B \in \mathcal{B}$. Using triangle inequality and symmetry from *Lemma 2.4*:

$$\forall B' \in \mathcal{B}, \mathsf{BagSep}(B, B, d) \leq \mathsf{BagSep}(B, B', d) + \mathsf{BagSep}(B', B, d)$$
$$\Rightarrow \quad \forall B' \in \mathcal{B}, \mathsf{BagSep}(B, B, d) \leq 2\mathsf{BagSep}(B', B, d)$$
$$\Rightarrow \quad \mathsf{BagSep}(B, B, d) \leq 2\frac{\Sigma_{B' \in \mathcal{B}, B' \neq B}\mathsf{BagSep}(B', B, d)}{|\mathcal{B}| - 1}$$
$$\Rightarrow \quad \mathsf{BagSep}(B, B, d) \leq 2\mathsf{InterBagSep}(B, d)$$
$$\Rightarrow \quad \mathsf{InterBagSep}(B, d)/\mathsf{BagSep}(B, B, d) \geq 1/2$$

$\square$

The squared euclidean distance is not a metric as it follows all properties other than the triangle inequality. Hence, we show the following

**Lemma A.1.** *For any $a, b \in R^n$, $\frac{1}{2}||a + b||_2^2 \leq ||a||_2^2 + ||b||_2^2$*

**Theorem A.2.** *Given $X$, $Y$ and $\mathcal{B}$, for any $B_1, B_2, B_3 \in \mathcal{B}$*
$\frac{1}{2}\mathsf{BagSep}(B_1, B_3, \ell_2^2) \leq \mathsf{BagSep}(B_1, B_2, \ell_2^2) + \mathsf{BagSep}(B_2, B_3, \ell_2^2)$

*Proof.* Follows by replacing triangle inequality in *Lemma 2.4* with inequality in *Lemma A.1* $\square$

**Corollary A.3.** $\mathsf{InterBagSep}(B, \ell_2^2)/\mathsf{BagSep}(B, B, \ell_2^2) \geq 1/4$

*Proof.* Follows by replacing inequality in proof of *Lemma 2.7* with inequality in *Theorem A.2* $\square$

## A.3 Proof of Corollary 2.8

*Proof.* Given $X$, $Y$ and $\mathcal{B}$, and metric $d$ in $R^n$. Starting with inequality in *Lemma 2.7*

$$\forall B \in \mathcal{B}, \mathsf{BagSep}(B, B, d) \leq 2\mathsf{InterBagSep}(B, d)$$
$$\Rightarrow \quad \Sigma_{B \in \mathcal{B}}\mathsf{BagSep}(B, B, d) \leq 2\Sigma_{B \in \mathcal{B}}\mathsf{InterBagSep}(B, d)$$
$$\Rightarrow \quad \frac{1}{|\mathcal{B}|}\mathsf{BagSep}(B, B, d) \leq 2\frac{1}{|\mathcal{B}|}\mathsf{InterBagSep}(B, d)$$
$$\Rightarrow \quad \mathsf{MeanInterBagSep}(\mathcal{B}, d)/\mathsf{MeanIntraBagSep}(\mathcal{B}, d) \geq 1/2$$

Starting with inequality for $\ell_2^2$-distance in *Lemma 2.7*, we get
$\mathsf{MeanInterBagSep}(\mathcal{B}, \ell_2^2)/\mathsf{MeanIntraBagSep}(\mathcal{B}, \ell_2^2) \geq 1/2$ $\square$

## A.4 Bag Distance Results using squared euclidean distance

We use the squared euclidean distance to compute the bag distances as it makes the computation faster. Algorithm 1 is used to compute the Bag Separation for any general metric $d$.

**Theorem A.4.** *Assuming the Bags to be disjoint, the running time of Algorithm 1 is $O(m^2n)$ where $m$ is the number of examples and $n$ is the dimension of the input space.*

*Proof.* Runtime $= \Sigma_{B_1 \in \mathcal{B}}\Sigma_{B_2 \in \mathcal{B}}|B_1||B_2|n = \Sigma_{B_1 \in \mathcal{B}}|B_1|\Sigma_{B_2 \in \mathcal{B}}|B_2|n = m^2n$ $\square$

Now, this computation can be simplified due to the following lemma.

**Lemma A.5.** *For any $B, B' \in \mathcal{B}$, $\mathsf{BagSep}(B, B', \ell_2^2) = \mathsf{AvgSqNorm}(B) + \mathsf{AvgSqNorm}(B') - 2\mathsf{DotProduct}(Mean(B), Mean(B'))$*

---

**Algorithm 1:** Compute Bag Separation of a dataset

---

**Data:** Set of bags $\mathcal{B}$, metric $d$ on $R^n$
**Result:** BagSepMatrix$(\mathcal{B}, d)$
BagSepMatrix $\leftarrow [0]_{|\mathcal{B}|x|\mathcal{B}|}$
**for** $B_1 \in \mathcal{B}$ **do**
    **for** $B_2 \in \mathcal{B}$ **do**
        **for** $i \in B_1$ **do**
            **for** $j \in B_2$ **do**
                BagSepMatrix$[B_1, B_2] \leftarrow$ BagSepMatrix$[B_1, B_2] + d(x^{(i)}, x^{(j)})$
            **end**
        **end**
        BagSepMatrix$[B_1, B_2] \leftarrow$ BagSepMatrix$[B_1, B_2]/(|B_1||B_2|)$
    **end**
**end**

---

*Proof.* Let $B = \{x_i | i \in [n]\}, B' = \{y_j | j \in [m]\}$

$$\mathsf{BagSep}(B, B', \ell_2^2) = \frac{1}{mn}\Sigma_{i=1}^{i=n}\Sigma_{j=1}^{j=m}||x_i - y_j||_2^2$$

$$\Rightarrow \quad \mathsf{BagSep}(B, B', \ell_2^2) = \frac{1}{n}\Sigma_{i=1}^{i=n}||x_i||_2^2 + \frac{1}{m}\Sigma_{j=1}^{j=m}||y_j||_2^2 - \frac{2}{mn}\Sigma_{i=1}^{i=n}\Sigma_{j=1}^{j=m}\langle x_i, y_j \rangle$$

$$\Rightarrow \quad \mathsf{BagSep}(B, B', \ell_2^2) = \frac{1}{n}\Sigma_{i=1}^{i=n}||x_i||_2^2 + \frac{1}{m}\Sigma_{j=1}^{j=m}||y_j||_2^2 - \frac{2}{mn}\langle \Sigma_{i=1}^{i=n}x_i, \Sigma_{j=1}^{j=m}y_j \rangle$$

$\square$

Algorithm 2 is used to compute the Bag Separation for squared euclidean distance.

---

**Algorithm 2:** Compute Bag Separation with squared euclidean distance

---

**Data:** Set of bags $\mathcal{B}$
**Result:** BagSepMatrix$(\mathcal{B}, \ell_2^2)$
BagSepMatrix $\leftarrow [0]_{|\mathcal{B}|x|\mathcal{B}|}$
AvgSqNorm $\leftarrow [0]_{|\mathcal{B}|}$
BagMeans $\leftarrow [0]_{|\mathcal{B}|xn}$
**for** $B \in \mathcal{B}$ **do**
    **for** $i \in B$ **do**
        AvgSqNorm$(B) \leftarrow$ AvgSqNorm$(B) + ||x^{(i)}||_2^2$
        BagMeans$(B) \leftarrow$ BagMeans$(B) + x^{(i)}$
    **end**
    AvgSqNorm$(B) \leftarrow$ AvgSqNorm$(B)/|B|$
    BagMeans$(B) \leftarrow$ BagMeans$(B)/|B|$
**end**
**for** $B_1 \in \mathcal{B}$ **do**
    **for** $B_2 \in \mathcal{B}$ **do**
        BagSepMatrix$[B_1, B_2] \leftarrow$
          AvgSqNorm$[B_1] +$ AvgSqNorm$[B_2] - 2$DotProduct(BagMeans$[B_1]$, Bagmeans$[B_2]$)
    **end**
**end**

---

**Theorem A.6.** *Assuming the Bags to be disjoint, the running time of Algorithm 2 is $O(mn+|\mathcal{B}|^2n+|\mathcal{B}|^2)$ where $m$ is the number of examples and n is the dimension of the input space.*

*Proof.* Runtime $= \Sigma_{B\in\mathcal{B}}|B|n + \Sigma_{B_1\in\mathcal{B}}\Sigma_{B_2\in\mathcal{B}}(1 + n) = mn + |\mathcal{B}|^2n + |\mathcal{B}|^2$ $\square$

A.5    ADVERSARIAL EXAMPLE OF BAGS WITH RATIO OF MEAN INTER TO INTRA BAG
       SEPARATION AS 1/2

Consider $X = \{x^{(1)}, x^{(2)}, x^{(3)}\}$ which lie on a straight line. The distances are as follows:

- $d(x^{(1)}, x^{(2)}) = d_1$
- $d(x^{(2)}, x^{(3)}) = d_2$
- $d(x^{(1)}, x^{(3)}) = d_1 + d_2$

We have two bags $B_1 = \{x^{(1)}, x^{(3)}\}$ and $B_2 = \{x^{(2)}\}$. The Intra-bag separations for both of them are as follows:

- $\mathsf{BagSep}(B_1, B_1, d) = \frac{1}{2^2}(d(x^{(1)}, x^{(1)}) + d(x^{(1)}, x^{(3)}) + d(x^{(3)}, x^{(1)}) + d(x^{(3)}, x^{(3)})) = \frac{1}{2}(d_1 + d_2)$
- $\mathsf{BagSep}(B_2, B_2, d) = 0$

Hence, $\mathsf{MeanIntraBagSep}(\mathcal{B}, d) = \frac{1}{4}(d_1 + d_2)$. Now, the bag separation between the bags is as follows:

- $\mathsf{BagSep}(B_1, B_2, d) = \frac{1}{1 \times 2}(d(x^{(1)}, x^{(2)}) + d(x^{(3)}, x^{(2)})) = \frac{1}{2}(d_1 + d_2)$
- $\mathsf{InterBagSep}(B_1, d) = \frac{1}{2-1}(\mathsf{BagSep}(B_1, B_2, d)) = \frac{1}{2}(d_1 + d_2)$
- $\mathsf{InterBagSep}(B_2, d) = \frac{1}{2-1}(\mathsf{BagSep}(B_2, B_1, d)) = \frac{1}{2}(d_1 + d_2)$

Hence, $\mathsf{MeanInterBagSep}(\mathcal{B}, d) = \frac{1}{2}(d_1 + d_2)$.
Hence, $\mathsf{MeanInterBagSep}(\mathcal{B}, d)/\mathsf{MeanIntraBagSep}(\mathcal{B}, d) = 1/2$

A.6    LLP MODEL TRAINING RESULTS

This table contains the AUC scores of all the experiments in Table 7. Each value represents the mean AUC score in percentage across 5 splits which are created as mentioned in 6.1. The error is the standard deviation of mean AUC scores across these 5 splits.

Table 8: AUC Scores obtained after different training configurations for all groupings

| Col1 | Col2 | E1 | E2 | E3 | E4 | E5 | E6 | E7 | E8 | E9 | E10 | E11 |
|---|---|---|---|---|---|---|---|---|---|---|---|---|
| C1 | C7 | 74.88±0.04 | 74.95±0.03 | 74.76±0.04 | 74.78±0.03 | 65.88±0.1 | 64.19±0.1 | 66.84±0.25 | 64.88±0.21 | 69.94±0.62 | 69.81±0.47 | 65.03±0.8 |
| C1 | C10 | 73.64±0.05 | 73.67±0.05 | 73.6±0.06 | 73.61±0.04 | 65.03±0.11 | 63.43±0.1 | 65.55±0.22 | 63.97±0.23 | 70.4±0.17 | 70.23±0.18 | 61.98±0.38 |
| C2 | C7 | 75.63±0.02 | 75.63±0.02 | 75.42±0.04 | 75.44±0.03 | 68.74±0.1 | 67.34±0.1 | 69.33±0.09 | 67.99±0.17 | 70.76±0.11 | 70.42±0.16 | 67.2±1.0 |
| C2 | C10 | 74.45±0.04 | 74.43±0.04 | 74.3±0.04 | 74.29±0.05 | 67.98±0.1 | 66.85±0.14 | 68.57±0.14 | 67.41±0.08 | 70.23±0.32 | 70.27±0.13 | 66.88±0.85 |
| C2 | C11 | 75.56±0.02 | 75.58±0.02 | 75.42±0.02 | 75.42±0.02 | 68.44±0.06 | 67.09±0.13 | 68.99±0.15 | 67.62±0.19 | 69.55±0.24 | 69.48±0.41 | 65.66±0.41 |
| C2 | C13 | 75.23±0.04 | 75.23±0.03 | 75.06±0.03 | 75.08±0.04 | 68.04±0.07 | 66.75±0.13 | 68.57±0.21 | 67.21±0.12 | 68.81±0.23 | 68.88±0.42 | 64.51±1.74 |
| C3 | C7 | 76.93±0.02 | 76.95±0.02 | 76.7±0.01 | 76.7±0.02 | 70.85±0.1 | 69.65±0.06 | 71.31±0.12 | 70.22±0.14 | 72.4±0.19 | 72.7±0.11 | 71.4±1.05 |
| C3 | C10 | 75.88±0.04 | 75.87±0.04 | 75.64±0.04 | 75.62±0.02 | 70.01±0.11 | 69.02±0.12 | 70.49±0.05 | 69.47±0.14 | 72.43±0.09 | 72.36±0.11 | 70.04±0.39 |
| C3 | C11 | 77.09±0.02 | 77.08±0.05 | 76.8±0.04 | 76.79±0.06 | 70.88±0.05 | 69.74±0.06 | 71.39±0.06 | 70.18±0.17 | 72.61±0.23 | 72.68±0.16 | 71.56±0.27 |
| C3 | C13 | 76.95±0.04 | 76.92±0.03 | 76.68±0.03 | 76.61±0.03 | 70.57±0.06 | 69.47±0.09 | 71.05±0.03 | 69.85±0.09 | 72.23±0.3 | 72.43±0.35 | 71.74±0.88 |
| C4 | C7 | 76.98±0.02 | 76.99±0.03 | 76.73±0.04 | 76.73±0.02 | 71.11±0.06 | 70.02±0.08 | 71.57±0.16 | 70.53±0.18 | 72.71±0.13 | 72.8±0.16 | 71.23±0.6 |
| C4 | C10 | 76.31±0.03 | 76.31±0.03 | 76.06±0.03 | 76.06±0.05 | 70.59±0.08 | 69.6±0.09 | 71.06±0.06 | 69.99±0.08 | 72.72±0.1 | 72.63±0.06 | 69.94±0.4 |
| C4 | C11 | 76.77±0.02 | 76.76±0.02 | 76.49±0.05 | 76.47±0.02 | 70.69±0.09 | 69.63±0.09 | 71.3±0.09 | 70.04±0.13 | 72.57±0.28 | 72.58±0.19 | 71.34±0.51 |
| C4 | C13 | 76.6±0.02 | 76.59±0.01 | 76.32±0.02 | 76.3±0.02 | 70.44±0.11 | 69.26±0.09 | 70.89±0.09 | 69.85±0.07 | 72.32±0.07 | 72.22±0.15 | 71.35±0.91 |
| C4 | C15 | 75.51±0.02 | 75.47±0.03 | 74.9±0.07 | 74.85±0.06 | 70.67±0.12 | 67.69±0.09 | 69.07±0.12 | 68.06±0.23 | 71.13±0.3 | 71.41±0.23 | 66.68±0.61 |
| C6 | C7 | 73.56±0.04 | 73.61±0.04 | 73.41±0.04 | 73.4±0.04 | 64.8±0.05 | 62.87±0.25 | 65.57±0.13 | 63.81±0.12 | 68.23±0.33 | 68.26±0.48 | 62.62±1.64 |
| C6 | C10 | 72.89±0.04 | 72.9±0.05 | 72.79±0.04 | 72.81±0.05 | 65.01±0.08 | 62.88±0.31 | 65.49±0.16 | 63.84±0.44 | 69.84±0.24 | 69.75±0.22 | 63.3±1.28 |
| C7 | C8 | 74.55±0.02 | 74.58±0.03 | 74.41±0.02 | 74.41±0.04 | 65.17±0.16 | 63.35±0.24 | 66.12±0.17 | 64.27±0.13 | 68.87±0.3 | 68.96±0.37 | 64.87±0.58 |
| C7 | C10 | 73.49±0.03 | 73.5±0.02 | 73.35±0.03 | 73.31±0.03 | 64.32±0.16 | 62.66±0.17 | 65.23±0.27 | 63.2±0.19 | 69.11±0.34 | 68.77±0.28 | 62.31±2.02 |
| C7 | C12 | 76.87±0.04 | 76.88±0.03 | 76.62±0.06 | 76.64±0.03 | 70.74±0.09 | 69.6±0.21 | 71.37±0.1 | 70.02±0.12 | 72.48±0.11 | 72.74±0.15 | 70.85±0.42 |
| C7 | C14 | 74.66±0.05 | 74.68±0.05 | 74.46±0.05 | 74.48±0.06 | 66.92±0.14 | 65.18±0.11 | 67.52±0.12 | 65.84±0.12 | 68.9±0.56 | 68.74±0.51 | 65.31±0.99 |
| C7 | C15 | 76.89±0.03 | 76.89±0.03 | 76.67±0.02 | 76.64±0.04 | 71.0±0.04 | 69.89±0.07 | 71.65±0.11 | 70.35±0.09 | 72.5±0.06 | 72.51±0.21 | 70.6±0.29 |
| C7 | C16 | 76.89±0.04 | 76.91±0.03 | 76.66±0.03 | 76.66±0.05 | 70.98±0.05 | 69.83±0.13 | 71.41±0.08 | 70.31±0.22 | 72.61±0.11 | 72.72±0.14 | 70.95±0.18 |
| C7 | C18 | 76.56±0.03 | 76.55±0.03 | 76.34±0.05 | 76.31±0.04 | 70.4±0.1 | 69.31±0.07 | 70.85±0.05 | 69.75±0.15 | 71.87±0.16 | 71.86±0.07 | 70.06±0.44 |
| C7 | C20 | 74.43±0.06 | 74.46±0.06 | 74.28±0.05 | 74.28±0.08 | 64.65±0.12 | 63.09±0.2 | 65.21±0.2 | 63.64±0.1 | 67.98±0.82 | 67.79±0.24 | 63.06±0.89 |
| C7 | C21 | 76.89±0.04 | 76.91±0.03 | 76.64±0.04 | 76.67±0.04 | 70.87±0.08 | 69.76±0.18 | 71.48±0.05 | 70.38±0.13 | 72.5±0.11 | 72.84±0.16 | 71.49±1.01 |
| C7 | C24 | 76.38±0.04 | 76.35±0.03 | 76.11±0.03 | 76.04±0.06 | 70.39±0.05 | 69.28±0.17 | 70.74±0.12 | 69.78±0.09 | 72.3±0.16 | 72.18±0.06 | 70.9±0.87 |
| C7 | C26 | 75.45±0.02 | 75.42±0.04 | 75.14±0.03 | 75.11±0.04 | 69.64±0.07 | 68.4±0.11 | 70.0±0.04 | 68.97±0.12 | 71.23±0.04 | 71.14±0.08 | 69.27±0.4 |
| C10 | C12 | 75.91±0.04 | 75.9±0.04 | 75.67±0.04 | 75.66±0.04 | 70.04±0.07 | 69.1±0.1 | 70.44±0.08 | 69.4±0.11 | 72.43±0.32 | 72.31±0.17 | 70.52±0.22 |
| C10 | C14 | 73.59±0.04 | 73.6±0.04 | 73.53±0.04 | 73.48±0.05 | 66.77±0.1 | 65.36±0.18 | 67.27±0.09 | 65.86±0.26 | 69.69±0.29 | 69.64±0.19 | 65.08±0.73 |
| C10 | C15 | 75.76±0.04 | 75.75±0.04 | 75.58±0.05 | 75.56±0.04 | 70.07±0.05 | 68.83±0.11 | 70.63±0.08 | 69.51±0.17 | 71.88±0.11 | 71.93±0.02 | 69.52±0.19 |
| C10 | C16 | 76.03±0.02 | 76.03±0.03 | 75.76±0.03 | 75.77±0.04 | 70.2±0.09 | 69.25±0.06 | 70.79±0.06 | 69.66±0.1 | 72.48±0.13 | 72.44±0.11 | 70.24±0.56 |

| | | | | | | | | | | | | |
|---|---|---|---|---|---|---|---|---|---|---|---|---|
| C10 | C17 | 73.18±0.04 | 73.18±0.04 | 73.12±0.04 | 73.06±0.02 | 63.66±0.13 | 62.33±0.21 | 64.55±0.14 | 62.78±0.18 | 68.94±0.19 | 69.23±0.3 | 62.02±1.2 |
| C10 | C18 | 75.29±0.03 | 75.28±0.03 | 75.12±0.03 | 75.09±0.05 | 69.24±0.1 | 68.29±0.09 | 69.8±0.07 | 68.66±0.09 | 71.26±0.19 | 71.21±0.14 | 68.77±0.45 |
| C10 | C20 | 73.41±0.04 | 73.4±0.04 | 73.36±0.04 | 73.29±0.05 | 65.07±0.19 | 63.46±0.22 | 65.75±0.18 | 64.13±0.24 | 69.35±0.52 | 69.53±0.29 | 62.98±0.61 |
| C10 | C21 | 76.02±0.03 | 76.01±0.04 | 75.77±0.05 | 75.76±0.04 | 70.11±0.09 | 69.23±0.09 | 70.7±0.09 | 69.62±0.2 | 72.38±0.08 | 72.46±0.09 | 70.54±0.67 |
| C10 | C24 | 75.4±0.03 | 75.37±0.01 | 75.18±0.02 | 75.11±0.04 | 69.72±0.04 | 68.83±0.08 | 70.12±0.09 | 69.24±0.1 | 72.35±0.1 | 71.95±0.14 | 69.32±0.51 |
| C10 | C26 | 74.73±0.03 | 74.72±0.03 | 74.48±0.06 | 74.46±0.05 | 69.05±0.11 | 67.94±0.08 | 69.43±0.06 | 68.44±0.15 | 71.03±0.26 | 70.78±0.25 | 68.03±0.97 |
| C11 | C12 | 76.98±0.05 | 76.98±0.03 | 76.71±0.04 | 76.72±0.03 | 70.72±0.09 | 69.57±0.1 | 71.37±0.14 | 69.96±0.15 | 72.59±0.13 | 72.65±0.13 | 71.47±0.18 |
| C11 | C15 | 76.85±0.02 | 76.83±0.01 | 76.61±0.01 | 76.66±0.04 | 70.76±0.08 | 69.68±0.08 | 71.37±0.1 | 70.1±0.09 | 72.39±0.11 | 72.32±0.16 | 71.1±0.42 |
| C11 | C16 | 76.87±0.03 | 76.86±0.04 | 76.6±0.02 | 76.59±0.03 | 70.81±0.08 | 69.65±0.06 | 71.39±0.03 | 70.08±0.06 | 72.69±0.3 | 72.51±0.11 | 71.53±0.58 |
| C11 | C18 | 76.54±0.05 | 76.55±0.06 | 76.33±0.05 | 76.33±0.04 | 70.24±0.09 | 69.05±0.11 | 70.68±0.11 | 69.46±0.09 | 71.69±0.17 | 71.78±0.14 | 70.3±0.78 |
| C11 | C21 | 77.0±0.02 | 76.99±0.01 | 76.73±0.02 | 76.72±0.02 | 70.92±0.09 | 69.73±0.11 | 71.48±0.1 | 70.28±0.16 | 72.71±0.17 | 72.78±0.18 | 72.04±1.24 |
| C11 | C24 | 76.31±0.03 | 76.27±0.03 | 76.01±0.04 | 75.93±0.04 | 70.15±0.07 | 69.09±0.08 | 70.57±0.07 | 69.46±0.09 | 72.07±0.14 | 72.15±0.19 | 70.98±0.44 |
| C11 | C26 | 75.89±0.03 | 75.86±0.04 | 75.6±0.05 | 75.58±0.04 | 69.85±0.07 | 68.67±0.29 | 70.37±0.11 | 69.25±0.09 | 71.24±0.16 | 71.42±0.1 | 69.18±0.39 |
| C12 | C13 | 76.83±0.04 | 76.82±0.05 | 76.55±0.05 | 76.54±0.04 | 70.4±0.05 | 69.41±0.06 | 71.0±0.08 | 69.71±0.1 | 72.01±0.16 | 72.38±0.07 | 71.26±0.42 |
| C13 | C15 | 76.82±0.02 | 76.8±0.02 | 76.61±0.03 | 76.57±0.04 | 70.58±0.06 | 69.44±0.15 | 71.03±0.12 | 69.91±0.1 | 71.99±0.11 | 72.04±0.2 | 70.7±0.3 |
| C13 | C16 | 76.71±0.02 | 76.71±0.02 | 76.44±0.05 | 76.42±0.02 | 70.51±0.07 | 69.42±0.09 | 71.06±0.14 | 69.86±0.15 | 72.36±0.21 | 72.24±0.12 | 72.07±0.53 |
| C13 | C18 | 76.33±0.03 | 76.32±0.04 | 76.11±0.02 | 76.11±0.04 | 69.99±0.1 | 68.8±0.11 | 70.41±0.07 | 69.17±0.1 | 71.18±0.13 | 71.38±0.19 | 68.88±0.66 |
| C13 | C21 | 76.83±0.02 | 76.83±0.02 | 76.57±0.02 | 76.56±0.02 | 70.57±0.03 | 69.47±0.09 | 71.06±0.15 | 69.82±0.07 | 72.32±0.12 | 72.46±0.14 | 71.5±0.23 |
| C13 | C24 | 76.13±0.03 | 76.09±0.02 | 75.81±0.02 | 75.8±0.02 | 69.89±0.02 | 68.91±0.07 | 70.2±0.06 | 69.29±0.12 | 71.94±0.11 | 71.77±0.13 | 70.35±0.41 |
| C13 | C26 | 75.96±0.02 | 75.92±0.03 | 75.68±0.05 | 75.59±0.04 | 69.85±0.07 | 68.47±0.15 | 70.29±0.05 | 69.13±0.19 | 71.53±0.14 | 71.34±0.19 | 69.16±0.61 |

## A.7 INSTANCE-LEVEL MODEL TRAINING RESULTS

We perform all the experiments mentioned in 7 on instance-level data. The process remains similar for different configurations as described in Sec. 6.1. We perform a train-test spilt of 80:20 on the dataset. We then train using instance level mini-batch gradient descent for the same number of epochs, using the same optimizer, model, learning rate schedule and the instance-level variant of the loss function. We again report the AUC scores as described in Sec. 6.1.

Table 9: AUC scores obtained by instance level training on Criteo

| Experiment | Model | Optimizer | Loss Function | AUC Score |
|---|---|---|---|---|
| E1 | 1-Layer MLP | Adam | BCE | 79.23 |
| E2 | 1-Layer MLP | Adam | MSE | 79.2 |
| E3 | 2-Layer MLP | Adam | BCE | 80.1 |
| E4 | 2-Layer MLP | Adam | MSE | 79.94 |
| E5 | 1-Layer MLP | SGD | BCE | 80.7 |
| E6 | 1-Layer MLP | SGD | MSE | 80.54 |
| E7 | 2-Layer MLP | SGD | BCE | 79.17 |
| E8 | 2-Layer MLP | SGD | MSE | 80.56 |
| E9 | AutoInt | Adam | BCE | 80.66 |
| E10 | AutoInt | Adam | MSE | 80.7 |
| E11 | AutoInt | Adam | MSE | 79.02 |

## A.8 BAG LEVEL STATISTICS

The bag level statistics for all 349 groupings is as follows. The Groupings which are emboldened pass our filters and are used for training.

Table 10: Bag Level Statistics of all the Groupings (Emboldened : Used for Training)

| Col1 | Col2 | No. Bags Created | No. of bags left after clipping | Percentage of Inst. left after clipping | Mean Bag size | Standard Deviation of Bag sizes |
|------|------|------------------|--------------------------------|----------------------------------------|---------------|--------------------------------|
| C1 | - | 1443 | 1261 | 0.7 | 256.16 | 386.61 |
| C3 | - | 175781 | 39052 | 18.42 | 216.17 | 316.89 |
| C4 | - | 128509 | 38802 | 21.68 | 256.11 | 359.9 |
| C7 | - | 11930 | 7839 | 12.39 | 724.69 | 636.25 |
| C8 | - | 629 | 531 | 0.34 | 295.24 | 446.88 |
| C10 | - | 41224 | 20252 | 17.01 | 384.92 | 482.64 |
| C11 | - | 5160 | 2519 | 4.57 | 832.04 | 683.76 |
| C12 | - | 174835 | 39444 | 18.79 | 218.32 | 318.82 |
| C13 | - | 3175 | 1221 | 2.98 | 1120.01 | 684.56 |
| C15 | - | 11254 | 6514 | 7.28 | 512.67 | 569 |
| C16 | - | 165206 | 40109 | 20.06 | 229.24 | 334.09 |
| C18 | - | 4605 | 2623 | 3.32 | 580.78 | 607.85 |
| C19 | - | 2017 | 1300 | 1.99 | 701.55 | 651.27 |
| C21 | - | 172322 | 39781 | 19.28 | 222.14 | 322.1 |
| C24 | - | 56456 | 21694 | 14.66 | 309.88 | 421.33 |
| C26 | - | 43356 | 17702 | 11.57 | 299.58 | 404.4 |
| C1 | C2 | 144029 | 11999 | 9.12 | 348.43 | 463.68 |
| C1 | C3 | 1986996 | 47251 | 22.36 | 216.96 | 318.6 |
| C1 | C4 | 1807068 | 58010 | 28.87 | 228.13 | 328.8 |
| C1 | C5 | 4852 | 4274 | 2.27 | 243.02 | 370.43 |
| C1 | C6 | 9950 | 2075 | 1.53 | 337.95 | 461.16 |
| **C1** | **C7** | **724000** | **55937** | **37.91** | **310.69** | **417.54** |
| C1 | C8 | 6577 | 5828 | 2.95 | 232.16 | 362.12 |
| C1 | C9 | 3005 | 1501 | 0.86 | 261.47 | 391.39 |
| **C1** | **C10** | **1034267** | **55528** | **32.44** | **267.81** | **374.19** |
| C1 | C11 | 421601 | 38117 | 27.5 | 330.67 | 441.68 |
| C1 | C12 | 1998608 | 48523 | 23.07 | 217.95 | 320.44 |
| C1 | C13 | 334925 | 32966 | 24.27 | 337.48 | 448.48 |
| C1 | C14 | 16449 | 2503 | 1.87 | 341.7 | 462.37 |
| C1 | C15 | 614530 | 39530 | 26.63 | 308.8 | 418.5 |
| C1 | C16 | 1992989 | 52671 | 25.52 | 222.14 | 324.99 |
| C1 | C17 | 12199 | 2245 | 1.7 | 346.25 | 460.77 |
| C1 | C18 | 365498 | 25570 | 18.07 | 324.03 | 435.59 |
| C1 | C19 | 144901 | 10476 | 7.18 | 314.3 | 423.13 |
| C1 | C20 | 5772 | 1732 | 1.26 | 332.59 | 460.3 |
| C1 | C21 | 2003740 | 50168 | 23.93 | 218.63 | 320.6 |
| C1 | C22 | 7269 | 1895 | 1.18 | 285.31 | 422.19 |
| C1 | C23 | 13043 | 2353 | 1.76 | 343.42 | 465.82 |
| C1 | C24 | 1062954 | 44816 | 24.81 | 253.74 | 359.8 |
| C1 | C25 | 23208 | 3126 | 2.34 | 342.44 | 458.17 |
| C1 | C26 | 816802 | 34315 | 18.37 | 245.35 | 349.16 |
| C2 | C3 | 645467 | 40900 | 19.63 | 219.98 | 321.84 |
| C2 | C4 | 588748 | 44692 | 24.03 | 246.51 | 351.32 |
| C2 | C5 | 41463 | 5606 | 4.9 | 400.34 | 505.1 |
| C2 | C6 | 4834 | 1907 | 2.85 | 684.3 | 653.76 |
| **C2** | **C7** | **444591** | **78261** | **48.74** | **285.51** | **377.68** |
| C2 | C8 | 73020 | 7884 | 6.42 | 373.27 | 480.22 |
| **C2** | **C10** | **945374** | **75614** | **41.23** | **249.98** | **345.32** |
| **C2** | **C11** | **221445** | **53308** | **39.09** | **336.11** | **431.53** |
| C2 | C12 | 649357 | 41375 | 19.97 | 221.23 | 322.15 |
| **C2** | **C13** | **162788** | **45206** | **34.87** | **353.59** | **446.07** |
| C2 | C14 | 2102 | 850 | 1.53 | 824.67 | 693.24 |
| C2 | C15 | 11390 | 6516 | 7.28 | 512.53 | 568.97 |
| C2 | C16 | 664651 | 43198 | 21.66 | 229.8 | 334.93 |
| C2 | C17 | 4855 | 2646 | 3.93 | 681.16 | 614.67 |
| C2 | C18 | 4631 | 2624 | 3.32 | 580.59 | 607.81 |
| C2 | C19 | 43411 | 10084 | 7.49 | 340.45 | 449.13 |
| C2 | C21 | 658369 | 42128 | 20.57 | 223.84 | 325.78 |
| C2 | C22 | 4337 | 1579 | 1.96 | 567.74 | 610.32 |
| C2 | C23 | 5080 | 2523 | 3.49 | 633.74 | 611.09 |
| C2 | C24 | 344266 | 31123 | 19.15 | 282.09 | 397.8 |
| C2 | C25 | 5228 | 1275 | 1.32 | 475.44 | 564.53 |
| C2 | C26 | 247348 | 24100 | 14.25 | 271.12 | 381.53 |
| C3 | C4 | 247003 | 55817 | 26.21 | 215.23 | 310.73 |
| C3 | C5 | 1114179 | 43934 | 20.88 | 217.81 | 318.81 |
| C3 | C6 | 844001 | 43641 | 20.62 | 216.6 | 316.98 |
| **C3** | **C7** | **7358757** | **80788** | **37.88** | **214.92** | **310.29** |
| C3 | C8 | 1438370 | 45282 | 21.51 | 217.78 | 319.66 |
| C3 | C9 | 309182 | 40293 | 19.2 | 218.42 | 320.38 |
| **C3** | **C10** | **7699949** | **71467** | **32** | **205.27** | **300.03** |
| **C3** | **C11** | **5826125** | **73331** | **35.78** | **223.64** | **324.41** |
| C3 | C12 | 187130 | 39997 | 18.82 | 215.65 | 316.49 |
| **C3** | **C13** | **5313650** | **70587** | **34.78** | **225.86** | **328.16** |
| C3 | C14 | 413784 | 40592 | 19.43 | 219.46 | 322.47 |
| C3 | C15 | 1076734 | 48427 | 26.22 | 248.23 | 363.96 |

| | | | | | | |
|---|---|---|---|---|---|---|
| C3 | C16 | 228945 | 48449 | 21.91 | 207.32 | 303.14 |
| C3 | C17 | 1082770 | 45864 | 22.1 | 220.88 | 325.63 |
| C3 | C18 | 726224 | 43469 | 22.2 | 234.16 | 346.86 |
| C3 | C19 | 483684 | 41188 | 19.99 | 222.47 | 327.54 |
| C3 | C20 | 346464 | 39600 | 18.8 | 217.58 | 316.56 |
| C3 | C21 | 205311 | 42537 | 19.54 | 210.52 | 309.43 |
| C3 | C22 | 561157 | 41869 | 19.95 | 218.46 | 323.34 |
| C3 | C23 | 898234 | 44976 | 21.41 | 218.17 | 318.12 |
| C3 | C24 | 229746 | 54447 | 27.12 | 228.34 | 329.83 |
| C3 | C25 | 352479 | 40796 | 19.45 | 218.52 | 318.79 |
| C3 | C26 | 406506 | 53753 | 26.8 | 228.57 | 329.43 |
| C4 | C5 | 955570 | 51305 | 26.41 | 235.97 | 336.72 |
| C4 | C6 | 649962 | 53352 | 27.1 | 232.83 | 332.09 |
| **C4** | **C7** | **7408443** | **103144** | **43.82** | **194.74** | **275.28** |
| C4 | C8 | 1264446 | 54197 | 27.55 | 233.03 | 334.98 |
| C4 | C9 | 230748 | 42003 | 22.99 | 250.9 | 353.24 |
| **C4** | **C10** | **8060081** | **86205** | **36.08** | **191.88** | **273.47** |
| **C4** | **C11** | **5628943** | **96009** | **44.42** | **212.1** | **299.69** |
| C4 | C12 | 245809 | 55816 | 26.26 | 215.67 | 311.09 |
| **C4** | **C13** | **5060557** | **93015** | **43.86** | **216.13** | **306.73** |
| C4 | C14 | 311943 | 47208 | 24.98 | 242.56 | 342.57 |
| **C4** | **C15** | **969150** | **56468** | **31.06** | **252.18** | **360.88** |
| C4 | C16 | 230280 | 49211 | 23.8 | 221.65 | 323.65 |
| C4 | C17 | 825745 | 56219 | 28.55 | 232.81 | 334.87 |
| C4 | C18 | 628441 | 47904 | 26.35 | 252.14 | 360.8 |
| C4 | C19 | 682269 | 44217 | 23.23 | 240.81 | 343.11 |
| C4 | C20 | 262150 | 43501 | 23.37 | 246.24 | 347.36 |
| C4 | C21 | 242560 | 54642 | 25.65 | 215.17 | 309.81 |
| C4 | C22 | 436550 | 46518 | 24.77 | 244.09 | 348.14 |
| C4 | C23 | 686712 | 54106 | 27.73 | 234.96 | 333.37 |
| C4 | C24 | 177482 | 45078 | 24.31 | 247.24 | 350.87 |
| C4 | C25 | 259638 | 43224 | 23.8 | 252.44 | 356.34 |
| C4 | C26 | 325526 | 49851 | 26.9 | 247.37 | 351.63 |
| C5 | C6 | 2260 | 579 | 0.51 | 401.17 | 511.77 |
| C5 | C7 | 290810 | 37354 | 29.42 | 360.98 | 463.2 |
| C5 | C8 | 2870 | 2486 | 1.44 | 265.84 | 406.85 |
| C5 | C10 | 480175 | 42678 | 27.35 | 293.75 | 398.83 |
| C5 | C11 | 159530 | 22921 | 19.1 | 381.99 | 488.02 |
| C5 | C12 | 1116548 | 44853 | 21.39 | 218.6 | 318.78 |
| C5 | C13 | 122621 | 18931 | 16.14 | 390.76 | 494.88 |
| C5 | C14 | 3934 | 780 | 0.68 | 399.88 | 504.78 |
| C5 | C15 | 235816 | 25697 | 19.44 | 346.74 | 454.78 |
| C5 | C16 | 1096716 | 47775 | 23.45 | 224.99 | 326.23 |
| C5 | C17 | 2619 | 649 | 0.57 | 403.67 | 512.2 |
| C5 | C18 | 128596 | 15061 | 12.02 | 366 | 472.77 |
| C5 | C19 | 53803 | 6342 | 5.04 | 364.25 | 467.44 |
| C5 | C21 | 1114287 | 46016 | 22.18 | 220.94 | 321.83 |
| C5 | C23 | 2904 | 698 | 0.62 | 404.87 | 524.76 |
| C5 | C24 | 517424 | 36088 | 21.14 | 268.51 | 375.4 |
| C5 | C25 | 5983 | 1066 | 0.91 | 390.25 | 492.66 |
| C5 | C26 | 401151 | 28221 | 16.01 | 260.12 | 365.26 |
| **C6** | **C7** | **82996** | **38449** | **35.84** | **427.31** | **492.33** |
| C6 | C8 | 4482 | 1006 | 0.82 | 371.25 | 477.59 |
| **C6** | **C10** | **235940** | **46981** | **31.8** | **310.24** | **411.03** |
| C6 | C11 | 38136 | 18249 | 21.92 | 550.51 | 575.83 |
| C6 | C12 | 840420 | 44685 | 21.2 | 217.45 | 318.9 |
| C6 | C13 | 25706 | 13586 | 17.92 | 604.75 | 595.34 |
| C6 | C15 | 70477 | 21774 | 19.65 | 413.78 | 509.4 |
| C6 | C16 | 808500 | 47935 | 23.52 | 224.89 | 324.94 |
| C6 | C18 | 31408 | 10996 | 10.79 | 449.9 | 531.49 |
| C6 | C19 | 14898 | 6625 | 6.02 | 416.53 | 489.81 |
| C6 | C21 | 833932 | 46083 | 21.94 | 218.28 | 317.68 |
| C6 | C24 | 301823 | 37787 | 22.33 | 270.9 | 377.4 |
| C6 | C26 | 238487 | 30115 | 17.26 | 262.65 | 372.55 |
| **C7** | **C8** | **436540** | **45042** | **33.37** | **339.57** | **446.24** |
| C7 | C9 | 11932 | 7842 | 12.4 | 724.75 | 636.39 |
| **C7** | **C10** | **238182** | **56575** | **40.52** | **328.31** | **428.64** |
| C7 | C11 | 17182 | 10090 | 14.47 | 657.49 | 620.35 |
| **C7** | **C12** | **7437839** | **84716** | **39.24** | **212.34** | **304.78** |
| C7 | C13 | 12348 | 7840 | 12.39 | 724.6 | 636.25 |
| **C7** | **C14** | **91450** | **38685** | **34.97** | **414.38** | **491.63** |
| **C7** | **C15** | **2885460** | **123209** | **53.78** | **200.1** | **280.57** |
| **C7** | **C16** | **7589473** | **94444** | **41.84** | **203.06** | **287.73** |
| C7 | C17 | 40660 | 21577 | 24.25 | 515.12 | 550.29 |
| **C7** | **C18** | **1770225** | **104806** | **51.31** | **224.44** | **314.1** |
| C7 | C19 | 1417980 | 41919 | 29.7 | 324.83 | 446.6 |
| **C7** | **C20** | **47371** | **30420** | **31.85** | **479.96** | **523.93** |
| **C7** | **C21** | **7515688** | **88970** | **40.43** | **208.31** | **297.62** |
| C7 | C22 | 70043 | 24036 | 23.42 | 446.7 | 517.49 |
| C7 | C23 | 69312 | 24818 | 23.79 | 439.47 | 519.91 |
| **C7** | **C24** | **5036272** | **103413** | **46.89** | **207.86** | **294.56** |
| C7 | C25 | 166138 | 26949 | 25.82 | 439.26 | 523.34 |
| **C7** | **C26** | **3804785** | **71088** | **35.56** | **229.28** | **335.78** |
| C8 | C9 | 1331 | 666 | 0.44 | 301.59 | 445.21 |
| C8 | C10 | 672470 | 48179 | 29.58 | 281.42 | 386.29 |
| C8 | C11 | 246919 | 28835 | 22.65 | 360.15 | 468.25 |

| | | | | | | |
|---|---|---|---|---|---|---|
| C8 | C12 | 1443335 | 46342 | 22.08 | 218.43 | 319.62 |
| C8 | C13 | 192815 | 24342 | 19.56 | 368.4 | 476.4 |
| C8 | C14 | 7603 | 1321 | 1.05 | 364.99 | 472.07 |
| C8 | C15 | 360202 | 31235 | 22.42 | 329.01 | 437.19 |
| C8 | C16 | 1426019 | 49809 | 24.34 | 224 | 325.98 |
| C8 | C17 | 5354 | 1163 | 0.95 | 374.7 | 489.44 |
| C8 | C18 | 204516 | 19086 | 14.39 | 345.67 | 453.35 |
| C8 | C19 | 83464 | 7972 | 5.88 | 338.4 | 444.15 |
| C8 | C20 | 2516 | 798 | 0.63 | 360.99 | 467.7 |
| C8 | C21 | 1442803 | 47665 | 22.87 | 219.92 | 320.34 |
| C8 | C22 | 3359 | 901 | 0.65 | 330.91 | 475.51 |
| C8 | C23 | 5847 | 1206 | 0.99 | 376.07 | 492.87 |
| C8 | C24 | 709532 | 39791 | 22.66 | 261.1 | 366.7 |
| C8 | C25 | 11146 | 1738 | 1.41 | 371.65 | 483.72 |
| C8 | C26 | 548366 | 30787 | 17.03 | 253.52 | 358.22 |
| C9 | C10 | 57586 | 23958 | 19.2 | 367.44 | 466.48 |
| C9 | C11 | 6317 | 3221 | 5.71 | 812.21 | 679.5 |
| C9 | C12 | 307826 | 40753 | 19.57 | 220.16 | 321.14 |
| C9 | C13 | 4085 | 1765 | 3.98 | 1033.66 | 690.28 |
| C9 | C15 | 22903 | 10022 | 10.06 | 459.94 | 537.52 |
| C9 | C16 | 293126 | 42011 | 20.92 | 228.26 | 332.12 |
| C9 | C18 | 9681 | 4547 | 5.06 | 509.69 | 566.6 |
| C9 | C19 | 4237 | 2214 | 2.6 | 538.05 | 582.85 |
| C9 | C21 | 304213 | 41192 | 20.05 | 223.11 | 322.92 |
| C9 | C24 | 104354 | 25184 | 16.51 | 300.58 | 411.1 |
| C9 | C26 | 79650 | 20247 | 12.76 | 288.85 | 394.88 |
| C10 | C11 | 91642 | 33364 | 27.97 | 384.27 | 480.8 |
| **C10** | **C12** | **7813021** | **73831** | **32.78** | **203.52** | **296.05** |
| C10 | C13 | 70927 | 29424 | 24.97 | 389.02 | 485.56 |
| **C10** | **C14** | **265775** | **47774** | **32.11** | **308.12** | **412** |
| **C10** | **C15** | **3893598** | **102841** | **44.66** | **199.07** | **283.28** |
| **C10** | **C16** | **8081277** | **79973** | **34.12** | **195.6** | **280.63** |
| **C10** | **C17** | **212092** | **44463** | **30.74** | **316.96** | **419.23** |
| **C10** | **C18** | **2524716** | **90181** | **42.31** | **215.05** | **305.56** |
| C10 | C19 | 1736492 | 48288 | 28.12 | 266.92 | 378.75 |
| **C10** | **C20** | **160915** | **41714** | **30** | **329.68** | **433.6** |
| **C10** | **C21** | **7935189** | **76455** | **33.32** | **199.78** | **288.52** |
| C10 | C22 | 169631 | 33712 | 24.05 | 327.08 | 430.99 |
| C10 | C23 | 196907 | 38464 | 27.12 | 323.17 | 423.14 |
| **C10** | **C24** | **5740783** | **87116** | **38.29** | **201.48** | **285.8** |
| C10 | C25 | 283759 | 37913 | 26.43 | 319.56 | 425.56 |
| **C10** | **C26** | **4449999** | **65658** | **31.38** | **219.1** | **319.86** |
| **C11** | **C12** | **5873140** | **76545** | **37.53** | **224.75** | **324.16** |
| C11 | C13 | 5460 | 2521 | 4.57 | 830.67 | 683.58 |
| C11 | C14 | 40358 | 19527 | 21.42 | 502.94 | 556.55 |
| **C11** | **C15** | **1847996** | **110386** | **53.39** | **221.71** | **306.51** |
| **C11** | **C16** | **5919824** | **85995** | **41.47** | **221.08** | **315.6** |
| C11 | C17 | 31753 | 16992 | 20.57 | 554.93 | 576.07 |
| **C11** | **C18** | **1088147** | **87394** | **47.51** | **249.22** | **344.12** |
| C11 | C19 | 947216 | 31892 | 20.29 | 291.63 | 426.76 |
| C11 | C20 | 20474 | 12946 | 18.01 | 637.83 | 604.71 |
| **C11** | **C21** | **5913565** | **80683** | **39.4** | **223.86** | **322.08** |
| C11 | C22 | 33530 | 12068 | 13.13 | 498.66 | 559.33 |
| C11 | C23 | 30567 | 13330 | 13.9 | 478.04 | 553.47 |
| **C11** | **C24** | **3654197** | **91578** | **45.3** | **226.75** | **319.09** |
| C11 | C25 | 91831 | 14952 | 14.43 | 442.27 | 545.98 |
| **C11** | **C26** | **2775118** | **63466** | **31.4** | **226.78** | **329.92** |
| **C12** | **C13** | **5350294** | **73654** | **36.56** | **227.52** | **329.06** |
| C12 | C14 | 410926 | 41573 | 19.85 | 218.85 | 320.3 |
| C12 | C15 | 1078982 | 49466 | 26.56 | 246.12 | 361.67 |
| C12 | C16 | 227478 | 48044 | 21.84 | 208.34 | 304.65 |
| C12 | C17 | 1078501 | 46908 | 22.61 | 220.91 | 324.81 |
| C12 | C18 | 728462 | 44143 | 22.46 | 233.28 | 344.19 |
| C12 | C19 | 489684 | 41677 | 20.36 | 223.94 | 329.07 |
| C12 | C20 | 346040 | 40127 | 19.11 | 218.34 | 316.26 |
| C12 | C21 | 201953 | 41802 | 19.51 | 213.92 | 313.62 |
| C12 | C22 | 559523 | 42652 | 20.45 | 219.76 | 324.27 |
| C12 | C23 | 894488 | 46019 | 21.95 | 218.66 | 318.19 |
| C12 | C24 | 228628 | 54766 | 27.35 | 228.93 | 329.94 |
| C12 | C25 | 351278 | 41326 | 19.84 | 220.07 | 319.52 |
| C12 | C26 | 406749 | 54177 | 27.06 | 229 | 329.34 |
| C13 | C14 | 26418 | 14778 | 17.58 | 545.39 | 576.63 |
| **C13** | **C15** | **1541997** | **103804** | **52.44** | **231.6** | **320.76** |
| **C13** | **C16** | **5369689** | **82839** | **40.64** | **224.87** | **321.63** |
| C13 | C17 | 21711 | 13065 | 17.24 | 605.06 | 594.38 |
| **C13** | **C18** | **891380** | **80044** | **45.09** | **258.23** | **354.54** |
| C13 | C19 | 814645 | 29681 | 18.26 | 282.07 | 416.78 |
| C13 | C20 | 12667 | 8879 | 14.33 | 740 | 625.94 |
| **C13** | **C21** | **5379799** | **77621** | **38.45** | **227.05** | **327.39** |
| C13 | C22 | 23085 | 8892 | 10.41 | 536.5 | 583.78 |
| C13 | C23 | 20835 | 10425 | 11.47 | 504.4 | 567.35 |
| **C13** | **C24** | **3231274** | **87527** | **44.17** | **231.32** | **326.29** |
| C13 | C25 | 69825 | 11776 | 11.75 | 457.51 | 559.89 |
| **C13** | **C26** | **2466154** | **61128** | **30.4** | **227.99** | **332.29** |
| C14 | C15 | 11281 | 6532 | 7.31 | 512.76 | 568.76 |
| C14 | C16 | 393993 | 44415 | 22.19 | 229.03 | 332.96 |

| | | | | | |
|---|---|---|---|---|---|
| C14 | C18 | 9799 | 5079 | 6.02 | 543.16 | 584.07 |
| C14 | C19 | 20339 | 6913 | 5.89 | 390.4 | 477.7 |
| C14 | C21 | 409726 | 42492 | 20.66 | 222.92 | 322.38 |
| C14 | C24 | 150980 | 30844 | 19.48 | 289.49 | 392.99 |
| C14 | C26 | 113197 | 23964 | 14.63 | 279.87 | 386.91 |
| C15 | C16 | 1092948 | 53050 | 28.67 | 247.75 | 359.24 |
| C15 | C17 | 90598 | 28410 | 24.63 | 397.42 | 493.15 |
| C15 | C18 | 12255 | 6521 | 7.29 | 512.29 | 568.84 |
| C15 | C19 | 186732 | 25897 | 18.35 | 324.82 | 434.28 |
| C15 | C20 | 12824 | 7268 | 8.37 | 528.03 | 581.62 |
| C15 | C21 | 1093512 | 50874 | 27.26 | 245.6 | 359.28 |
| C15 | C22 | 56802 | 15626 | 14.28 | 418.99 | 513.2 |
| C15 | C23 | 79514 | 22941 | 19.83 | 396.22 | 489.02 |
| C15 | C24 | 591651 | 43783 | 27.07 | 283.46 | 394.29 |
| C15 | C25 | 29235 | 9819 | 9.98 | 466.04 | 541.56 |
| C15 | C26 | 431970 | 36836 | 23.9 | 297.37 | 411.98 |
| C16 | C17 | 1028310 | 50610 | 24.88 | 225.31 | 328.36 |
| C16 | C18 | 727988 | 46323 | 24.31 | 240.57 | 350.21 |
| C16 | C19 | 592668 | 42843 | 21.45 | 229.46 | 335.21 |
| C16 | C20 | 331688 | 42038 | 20.78 | 226.66 | 327.87 |
| C16 | C21 | 220892 | 45463 | 21.16 | 213.33 | 311.16 |
| C16 | C22 | 535479 | 44618 | 22.28 | 228.89 | 334.42 |
| C16 | C23 | 852691 | 49312 | 24.22 | 225.11 | 323.37 |
| C16 | C24 | 218175 | 53197 | 26.54 | 228.67 | 330.42 |
| C16 | C25 | 337787 | 42815 | 21.46 | 229.76 | 332.85 |
| C16 | C26 | 388938 | 53940 | 27.25 | 231.58 | 334.5 |
| C17 | C18 | 38523 | 14374 | 13.84 | 441.28 | 518.96 |
| C17 | C19 | 17150 | 7252 | 5.98 | 378.04 | 462.68 |
| C17 | C21 | 1065077 | 48331 | 23.37 | 221.64 | 325.22 |
| C17 | C24 | 383364 | 41263 | 24.18 | 268.65 | 376.64 |
| C17 | C26 | 293309 | 31763 | 17.97 | 259.39 | 366.12 |
| C18 | C19 | 59982 | 13650 | 11.21 | 376.48 | 484.6 |
| C18 | C20 | 5016 | 2769 | 3.57 | 590.95 | 613.21 |
| C18 | C21 | 734247 | 45194 | 23.09 | 234.21 | 343.82 |
| C18 | C22 | 25833 | 7794 | 7.63 | 448.62 | 531.88 |
| C18 | C23 | 36318 | 12199 | 11.79 | 442.94 | 518.85 |
| C18 | C24 | 361722 | 33658 | 21.14 | 287.85 | 403.88 |
| C18 | C25 | 11302 | 3962 | 4.47 | 516.88 | 575.34 |
| C18 | C26 | 252997 | 26993 | 17.63 | 299.37 | 414.88 |
| C19 | C20 | 5946 | 3305 | 3.68 | 510.17 | 546.43 |
| C19 | C21 | 515626 | 42198 | 20.78 | 225.76 | 329.07 |
| C19 | C22 | 12243 | 3893 | 3.69 | 434.23 | 512.33 |
| C19 | C23 | 18121 | 6530 | 5.64 | 395.95 | 478.38 |
| C19 | C24 | 513765 | 30953 | 18.5 | 273.92 | 386.6 |
| C19 | C25 | 31248 | 7260 | 5.92 | 374.09 | 483.47 |
| C19 | C26 | 519124 | 25569 | 14.06 | 252.04 | 357.66 |
| C20 | C21 | 342773 | 40942 | 19.74 | 220.99 | 319.39 |
| C20 | C24 | 122667 | 27766 | 17.58 | 290.31 | 397.66 |
| C20 | C26 | 79271 | 21542 | 13.37 | 284.52 | 390.22 |
| C21 | C22 | 553344 | 43555 | 21.03 | 221.35 | 324.7 |
| C21 | C23 | 885098 | 47266 | 22.74 | 220.57 | 319.3 |
| C21 | C24 | 225892 | 54751 | 27.33 | 228.84 | 327.83 |
| C21 | C25 | 348750 | 42006 | 20.41 | 222.77 | 322.11 |
| C21 | C26 | 403280 | 54611 | 27.32 | 229.35 | 329.42 |
| C22 | C24 | 207779 | 29960 | 18.7 | 286.2 | 396.64 |
| C22 | C26 | 162373 | 23738 | 14.33 | 276.8 | 383.56 |
| C23 | C24 | 326117 | 38494 | 22.7 | 270.28 | 373.29 |
| C23 | C26 | 246190 | 29727 | 17.06 | 263.12 | 365.1 |
| C24 | C25 | 141933 | 26882 | 17.68 | 301.46 | 413.97 |
| C24 | C26 | 193330 | 30432 | 20.05 | 302.07 | 412.59 |
| C25 | C26 | 96107 | 21524 | 13.86 | 295.26 | 402.75 |
| C2 | - | 554 | 130 | 0.29 | 1012.17 | 693.33 |
| C5 | - | 305 | 238 | 0.17 | 328.15 | 449.85 |
| C6 | - | 19 | 5 | 0.01 | 918.6 | 709.03 |
| C14 | - | 27 | 2 | 0 | 1095 | 562 |
| C17 | - | 10 | 1 | 0 | 1292 | 0 |
| C22 | - | 18 | 5 | 0 | 193.4 | 155.81 |
| C23 | - | 15 | 1 | 0 | 62 | 0 |
| C25 | - | 86 | 24 | 0.02 | 304.54 | 269.11 |
| C2 | C9 | 1287 | 452 | 0.76 | 766.81 | 682.5 |
| C2 | C20 | 787 | 189 | 0.37 | 904.18 | 672.73 |
| C5 | C9 | 654 | 312 | 0.24 | 349.6 | 474.05 |
| C5 | C20 | 1220 | 441 | 0.39 | 403.24 | 526.74 |
| C5 | C22 | 1713 | 471 | 0.37 | 360.37 | 478.7 |
| C6 | C9 | 48 | 16 | 0.02 | 580.5 | 548.02 |
| C6 | C14 | 304 | 89 | 0.11 | 579.75 | 678.32 |
| C6 | C17 | 164 | 44 | 0.03 | 317.89 | 332.98 |
| C6 | C20 | 75 | 24 | 0.03 | 534.54 | 546.47 |
| C6 | C22 | 111 | 27 | 0.04 | 593.22 | 587.68 |
| C6 | C23 | 208 | 50 | 0.06 | 528.42 | 527.49 |
| C6 | C25 | 711 | 186 | 0.26 | 631.13 | 649.91 |
| C9 | C14 | 68 | 18 | 0.03 | 790.28 | 749.69 |
| C9 | C17 | 28 | 10 | 0.02 | 893.8 | 730.07 |
| C9 | C20 | 12 | 3 | 0.01 | 1145.33 | 535.07 |
| C9 | C22 | 46 | 10 | 0.01 | 548.9 | 454.85 |
| C9 | C23 | 40 | 11 | 0.02 | 672.18 | 709.77 |

| | | | | | |
|---|---|---|---|---|---|
| C9 | C25 | 206 | 57 | 0.07 | 580.17 | 655.06 |
| C14 | C17 | 238 | 79 | 0.17 | 966.99 | 656.28 |
| C14 | C20 | 80 | 13 | 0.02 | 624.23 | 582.95 |
| C14 | C22 | 294 | 96 | 0.13 | 641.21 | 680.65 |
| C14 | C23 | 197 | 61 | 0.11 | 827.28 | 746.54 |
| C14 | C25 | 744 | 214 | 0.27 | 577.49 | 595 |
| C17 | C20 | 40 | 4 | 0 | 323 | 128.08 |
| C17 | C22 | 153 | 34 | 0.04 | 530.21 | 477.38 |
| C17 | C23 | 135 | 23 | 0.02 | 480.09 | 573.73 |
| C17 | C25 | 719 | 194 | 0.31 | 738.04 | 703.07 |
| C20 | C22 | 67 | 13 | 0.03 | 899.38 | 755.05 |
| C20 | C23 | 55 | 5 | 0.01 | 1095.6 | 398.26 |
| C20 | C25 | 251 | 61 | 0.09 | 648.51 | 730.95 |
| C22 | C23 | 192 | 46 | 0.06 | 634.89 | 603.13 |
| C22 | C25 | 592 | 174 | 0.2 | 522.01 | 557.59 |
| C23 | C25 | 809 | 210 | 0.28 | 614.03 | 636.39 |

## A.9 BAG SIZES TO ACHIEVE 50, 70, 85, 95 PERCENTILE OF BAGS

This table contains the threshold bags sizes such that $t\%$ of the bags have at most that size, for $t = 50, 70, 85, 95$ for all $349$ groupings (removing groupings which were left with no bags after clipping). We perform K-Means of $308$ of these groupings that have more than $500$ bags after clipping. The cluster assigned to each bag is also listed.

Table 11: Threshold bag size values below which 50, 70, 85, 95% of bags present and clustering based on this distribution

| Col1 | Col2 | Bag size below which 50% bags | Bag size below which 70% bags | Bag size below which 85% bags | Bag size below which 95% bags | Clusters assigned on bag size distribution |
|---|---|---|---|---|---|---|
| C1 | - | 70 | 210 | 420 | 1130 | Very Short-tailed |
| C3 | - | 101 | 172 | 335 | 812 | Very Short-tailed |
| C4 | - | 117 | 212 | 430 | 1019 | Very Short-tailed |
| C7 | - | 502 | 951 | 1499 | 2076 | Very Long-tailed |
| C8 | - | 110 | 210 | 510 | 1383 | Short-tailed |
| C10 | - | 175 | 365 | 764 | 1533 | Short-tailed |
| C11 | - | 637 | 1155 | 1699 | 2194 | Very Long-tailed |
| C12 | - | 103 | 175 | 341 | 821 | Very Short-tailed |
| C13 | - | 1044 | 1498 | 1976 | 2349 | Very Long-tailed |
| C15 | - | 264 | 564 | 1121 | 1829 | Long-tailed |
| C16 | - | 106 | 184 | 364 | 890 | Very Short-tailed |
| C18 | - | 327 | 697 | 1271 | 1983 | Long-tailed |
| C19 | - | 456 | 917 | 1454 | 2153 | Very Long-tailed |
| C21 | - | 104 | 180 | 352 | 835 | Very Short-tailed |
| C24 | - | 136 | 264 | 559 | 1279 | Short-tailed |
| C26 | - | 136 | 256 | 538 | 1203 | Short-tailed |
| C1 | C2 | 151 | 308 | 671 | 1455 | Short-tailed |
| C1 | C3 | 103 | 171 | 335 | 825 | Very Short-tailed |
| C1 | C4 | 108 | 184 | 361 | 872 | Very Short-tailed |
| C1 | C5 | 70 | 190 | 415 | 1050 | Very Short-tailed |
| C1 | C6 | 142 | 286 | 647 | 1449 | Short-tailed |
| C1 | C7 | 140 | 271 | 569 | 1270 | Short-tailed |
| C1 | C8 | 70 | 140 | 361 | 1010 | Very Short-tailed |
| C1 | C9 | 91 | 197 | 449 | 1168 | Very Short-tailed |
| C1 | C10 | 122 | 224 | 454 | 1068 | Very Short-tailed |
| C1 | C11 | 145 | 290 | 624 | 1374 | Short-tailed |
| C1 | C12 | 103 | 172 | 338 | 829 | Very Short-tailed |
| C1 | C13 | 148 | 299 | 643 | 1397 | Short-tailed |
| C1 | C14 | 142 | 289 | 633 | 1461 | Short-tailed |
| C1 | C15 | 137 | 265 | 566 | 1267 | Short-tailed |
| C1 | C16 | 104 | 176 | 347 | 854 | Very Short-tailed |
| C1 | C17 | 151 | 302 | 655 | 1443 | Short-tailed |
| C1 | C18 | 142 | 281 | 606 | 1333 | Short-tailed |
| C1 | C19 | 140 | 272 | 575 | 1292 | Short-tailed |
| C1 | C20 | 138 | 277 | 645 | 1419 | Short-tailed |
| C1 | C21 | 103 | 174 | 339 | 830 | Very Short-tailed |
| C1 | C22 | 109 | 224 | 525 | 1247 | Short-tailed |
| C1 | C23 | 142 | 293 | 652 | 1456 | Short-tailed |
| C1 | C24 | 116 | 209 | 422 | 1006 | Very Short-tailed |
| C1 | C25 | 148 | 297 | 658 | 1417 | Short-tailed |
| C1 | C26 | 114 | 200 | 404 | 964 | Very Short-tailed |
| C2 | C3 | 102 | 176 | 344 | 837 | Very Short-tailed |
| C2 | C4 | 112 | 201 | 404 | 984 | Very Short-tailed |
| C2 | C5 | 175 | 377 | 816 | 1622 | Short-tailed |
| C2 | C6 | 419 | 878 | 1499 | 2112 | Very Long-tailed |
| C2 | C7 | 136 | 253 | 503 | 1102 | Very Short-tailed |
| C2 | C8 | 164 | 344 | 742 | 1523 | Short-tailed |
| C2 | C10 | 119 | 210 | 420 | 957 | Very Short-tailed |
| C2 | C11 | 157 | 311 | 637 | 1329 | Short-tailed |
| C2 | C12 | 103 | 177 | 347 | 842 | Very Short-tailed |
| C2 | C13 | 165 | 334 | 688 | 1398 | Short-tailed |

| | | | | | | |
|---|---|---|---|---|---|---|
| C2 | C14 | 613 | 1139 | 1701 | 2229 | Very Long-tailed |
| C2 | C15 | 264 | 564 | 1121 | 1829 | Long-tailed |
| C2 | C16 | 105 | 183 | 364 | 903 | Very Short-tailed |
| C2 | C17 | 479 | 855 | 1413 | 2010 | Very Long-tailed |
| C2 | C18 | 327 | 696 | 1271 | 1983 | Long-tailed |
| C2 | C19 | 151 | 304 | 630 | 1413 | Short-tailed |
| C2 | C21 | 104 | 180 | 353 | 853 | Very Short-tailed |
| C2 | C22 | 304 | 664 | 1245 | 2019 | Long-tailed |
| C2 | C23 | 395 | 788 | 1351 | 1992 | Long-tailed |
| C2 | C24 | 124 | 232 | 484 | 1171 | Very Short-tailed |
| C2 | C25 | 210 | 506 | 1041 | 1813 | Long-tailed |
| C2 | C26 | 122 | 224 | 458 | 1089 | Very Short-tailed |
| C3 | C4 | 103 | 175 | 335 | 803 | Very Short-tailed |
| C3 | C5 | 103 | 174 | 336 | 819 | Very Short-tailed |
| C3 | C6 | 102 | 170 | 334 | 837 | Very Short-tailed |
| C3 | C7 | 103 | 173 | 334 | 804 | Very Short-tailed |
| C3 | C8 | 103 | 173 | 336 | 832 | Very Short-tailed |
| C3 | C9 | 102 | 175 | 341 | 827 | Very Short-tailed |
| C3 | C10 | 100 | 164 | 312 | 753 | Very Short-tailed |
| C3 | C11 | 105 | 179 | 351 | 853 | Very Short-tailed |
| C3 | C12 | 101 | 172 | 334 | 810 | Very Short-tailed |
| C3 | C13 | 106 | 180 | 357 | 864 | Very Short-tailed |
| C3 | C14 | 103 | 175 | 337 | 836 | Very Short-tailed |
| C3 | C15 | 109 | 193 | 402 | 1030 | Very Short-tailed |
| C3 | C16 | 100 | 166 | 316 | 764 | Very Short-tailed |
| C3 | C17 | 103 | 175 | 344 | 846 | Very Short-tailed |
| C3 | C18 | 104 | 183 | 371 | 950 | Very Short-tailed |
| C3 | C19 | 102 | 175 | 350 | 858 | Very Short-tailed |
| C3 | C20 | 103 | 175 | 340 | 820 | Very Short-tailed |
| C3 | C21 | 100 | 167 | 322 | 786 | Very Short-tailed |
| C3 | C22 | 102 | 172 | 338 | 837 | Very Short-tailed |
| C3 | C23 | 103 | 174 | 340 | 834 | Very Short-tailed |
| C3 | C24 | 107 | 185 | 364 | 875 | Very Short-tailed |
| C3 | C25 | 103 | 175 | 339 | 826 | Very Short-tailed |
| C3 | C26 | 107 | 185 | 365 | 877 | Very Short-tailed |
| C4 | C5 | 110 | 192 | 382 | 912 | Very Short-tailed |
| C4 | C6 | 110 | 190 | 373 | 895 | Very Short-tailed |
| C4 | C7 | 100 | 161 | 296 | 685 | Very Short-tailed |
| C4 | C8 | 109 | 188 | 372 | 895 | Very Short-tailed |
| C4 | C9 | 116 | 207 | 418 | 987 | Very Short-tailed |
| C4 | C10 | 97 | 157 | 290 | 669 | Very Short-tailed |
| C4 | C11 | 104 | 174 | 333 | 780 | Very Short-tailed |
| C4 | C12 | 103 | 175 | 337 | 804 | Very Short-tailed |
| C4 | C13 | 106 | 177 | 341 | 800 | Very Short-tailed |
| C4 | C14 | 114 | 200 | 397 | 945 | Very Short-tailed |
| C4 | C15 | 114 | 203 | 417 | 1030 | Very Short-tailed |
| C4 | C16 | 104 | 177 | 349 | 843 | Very Short-tailed |
| C4 | C17 | 110 | 189 | 374 | 892 | Very Short-tailed |
| C4 | C18 | 114 | 205 | 413 | 1020 | Very Short-tailed |
| C4 | C19 | 111 | 196 | 392 | 953 | Very Short-tailed |
| C4 | C20 | 114 | 202 | 407 | 965 | Very Short-tailed |
| C4 | C21 | 103 | 175 | 337 | 793 | Very Short-tailed |
| C4 | C22 | 113 | 199 | 400 | 964 | Very Short-tailed |
| C4 | C23 | 110 | 192 | 384 | 904 | Very Short-tailed |
| C4 | C24 | 113 | 202 | 408 | 976 | Very Short-tailed |
| C4 | C25 | 116 | 208 | 418 | 1012 | Very Short-tailed |
| C4 | C26 | 114 | 202 | 406 | 983 | Very Short-tailed |
| C5 | C6 | 176 | 358 | 821 | 1641 | Short-tailed |
| C5 | C7 | 163 | 331 | 702 | 1465 | Short-tailed |
| C5 | C8 | 70 | 210 | 470 | 1210 | Very Short-tailed |
| C5 | C10 | 133 | 253 | 523 | 1192 | Short-tailed |
| C5 | C11 | 169 | 356 | 765 | 1556 | Short-tailed |
| C5 | C12 | 103 | 174 | 338 | 825 | Very Short-tailed |
| C5 | C13 | 173 | 368 | 791 | 1582 | Short-tailed |
| C5 | C14 | 179 | 373 | 825 | 1638 | Short-tailed |
| C5 | C15 | 153 | 310 | 667 | 1416 | Short-tailed |
| C5 | C16 | 105 | 180 | 355 | 860 | Very Short-tailed |
| C5 | C17 | 180 | 392 | 789 | 1699 | Short-tailed |
| C5 | C18 | 162 | 334 | 720 | 1498 | Short-tailed |
| C5 | C19 | 162 | 339 | 712 | 1506 | Short-tailed |
| C5 | C21 | 104 | 177 | 344 | 833 | Very Short-tailed |
| C5 | C23 | 174 | 367 | 821 | 1705 | Short-tailed |
| C5 | C24 | 122 | 223 | 459 | 1093 | Very Short-tailed |
| C5 | C25 | 173 | 372 | 772 | 1589 | Short-tailed |
| C5 | C26 | 119 | 215 | 440 | 1034 | Very Short-tailed |
| C6 | C7 | 222 | 443 | 854 | 1600 | Short-tailed |
| C6 | C8 | 165 | 343 | 729 | 1534 | Short-tailed |
| C6 | C10 | 144 | 271 | 567 | 1246 | Short-tailed |
| C6 | C11 | 310 | 640 | 1172 | 1872 | Long-tailed |
| C6 | C12 | 102 | 171 | 335 | 839 | Very Short-tailed |
| C6 | C13 | 367 | 728 | 1274 | 1944 | Long-tailed |
| C6 | C15 | 190 | 400 | 837 | 1645 | Short-tailed |
| C6 | C16 | 106 | 180 | 355 | 872 | Very Short-tailed |
| C6 | C18 | 217 | 464 | 937 | 1730 | Long-tailed |
| C6 | C19 | 211 | 422 | 822 | 1578 | Short-tailed |
| C6 | C21 | 103 | 173 | 338 | 840 | Very Short-tailed |

| | | | | | | |
|---|---|---|---|---|---|---|
| C6 | C24 | 124 | 228 | 459 | 1091 | Very Short-tailed |
| C6 | C26 | 119 | 216 | 436 | 1066 | Very Short-tailed |
| C7 | C8 | 153 | 303 | 646 | 1383 | Short-tailed |
| C7 | C9 | 502 | 951 | 1499 | 2076 | Very Long-tailed |
| C7 | C10 | 151 | 296 | 613 | 1316 | Short-tailed |
| C7 | C11 | 419 | 832 | 1392 | 2019 | Very Long-tailed |
| C7 | C12 | 103 | 173 | 329 | 787 | Very Short-tailed |
| C7 | C13 | 502 | 951 | 1499 | 2076 | Very Long-tailed |
| C7 | C14 | 205 | 417 | 833 | 1577 | Short-tailed |
| C7 | C15 | 103 | 168 | 306 | 696 | Very Short-tailed |
| C7 | C16 | 102 | 168 | 311 | 727 | Very Short-tailed |
| C7 | C17 | 284 | 581 | 1076 | 1791 | Long-tailed |
| C7 | C18 | 110 | 188 | 360 | 830 | Very Short-tailed |
| C7 | C19 | 133 | 273 | 619 | 1384 | Short-tailed |
| C7 | C20 | 261 | 523 | 977 | 1703 | Long-tailed |
| C7 | C21 | 103 | 171 | 321 | 761 | Very Short-tailed |
| C7 | C22 | 223 | 467 | 929 | 1670 | Long-tailed |
| C7 | C23 | 210 | 450 | 918 | 1676 | Long-tailed |
| C7 | C24 | 104 | 172 | 322 | 748 | Very Short-tailed |
| C7 | C25 | 209 | 450 | 916 | 1700 | Long-tailed |
| C7 | C26 | 106 | 181 | 364 | 887 | Very Short-tailed |
| C8 | C9 | 121 | 220 | 596 | 1443 | Short-tailed |
| C8 | C10 | 128 | 238 | 491 | 1132 | Very Short-tailed |
| C8 | C11 | 159 | 326 | 705 | 1486 | Short-tailed |
| C8 | C12 | 104 | 174 | 339 | 833 | Very Short-tailed |
| C8 | C13 | 162 | 337 | 727 | 1516 | Short-tailed |
| C8 | C14 | 156 | 348 | 719 | 1505 | Short-tailed |
| C8 | C15 | 146 | 288 | 619 | 1350 | Short-tailed |
| C8 | C16 | 105 | 179 | 352 | 860 | Very Short-tailed |
| C8 | C17 | 158 | 344 | 748 | 1552 | Short-tailed |
| C8 | C18 | 154 | 308 | 667 | 1422 | Short-tailed |
| C8 | C19 | 150 | 306 | 646 | 1381 | Short-tailed |
| C8 | C20 | 161 | 330 | 754 | 1537 | Short-tailed |
| C8 | C21 | 104 | 176 | 342 | 839 | Very Short-tailed |
| C8 | C22 | 121 | 268 | 660 | 1473 | Short-tailed |
| C8 | C23 | 160 | 343 | 737 | 1580 | Short-tailed |
| C8 | C24 | 119 | 216 | 441 | 1049 | Very Short-tailed |
| C8 | C25 | 158 | 334 | 743 | 1529 | Short-tailed |
| C8 | C26 | 116 | 209 | 426 | 998 | Very Short-tailed |
| C9 | C10 | 168 | 344 | 709 | 1468 | Short-tailed |
| C9 | C11 | 610 | 1119 | 1671 | 2185 | Very Long-tailed |
| C9 | C12 | 103 | 177 | 346 | 834 | Very Short-tailed |
| C9 | C13 | 925 | 1394 | 1903 | 2298 | Very Long-tailed |
| C9 | C15 | 223 | 482 | 966 | 1749 | Long-tailed |
| C9 | C16 | 106 | 183 | 364 | 890 | Very Short-tailed |
| C9 | C18 | 264 | 560 | 1079 | 1863 | Long-tailed |
| C9 | C19 | 284 | 620 | 1162 | 1923 | Long-tailed |
| C9 | C21 | 105 | 181 | 355 | 845 | Very Short-tailed |
| C9 | C24 | 133 | 256 | 538 | 1245 | Short-tailed |
| C9 | C26 | 130 | 244 | 512 | 1156 | Very Short-tailed |
| C10 | C11 | 177 | 365 | 763 | 1523 | Short-tailed |
| C10 | C12 | 99 | 163 | 309 | 744 | Very Short-tailed |
| C10 | C13 | 177 | 372 | 777 | 1548 | Short-tailed |
| C10 | C14 | 141 | 272 | 557 | 1233 | Short-tailed |
| C10 | C15 | 101 | 164 | 303 | 705 | Very Short-tailed |
| C10 | C16 | 98 | 158 | 296 | 697 | Very Short-tailed |
| C10 | C17 | 145 | 280 | 581 | 1281 | Short-tailed |
| C10 | C18 | 105 | 176 | 338 | 786 | Very Short-tailed |
| C10 | C19 | 118 | 218 | 454 | 1081 | Very Short-tailed |
| C10 | C20 | 151 | 295 | 614 | 1342 | Short-tailed |
| C10 | C21 | 99 | 161 | 302 | 721 | Very Short-tailed |
| C10 | C22 | 147 | 290 | 614 | 1336 | Short-tailed |
| C10 | C23 | 149 | 289 | 595 | 1300 | Short-tailed |
| C10 | C24 | 100 | 165 | 310 | 728 | Very Short-tailed |
| C10 | C25 | 144 | 281 | 589 | 1294 | Short-tailed |
| C10 | C26 | 104 | 176 | 341 | 833 | Very Short-tailed |
| C11 | C12 | 106 | 181 | 356 | 857 | Very Short-tailed |
| C11 | C13 | 636 | 1151 | 1698 | 2192 | Very Long-tailed |
| C11 | C14 | 265 | 545 | 1067 | 1810 | Long-tailed |
| C11 | C15 | 110 | 188 | 355 | 810 | Very Short-tailed |
| C11 | C16 | 106 | 180 | 349 | 830 | Very Short-tailed |
| C11 | C17 | 312 | 647 | 1181 | 1872 | Long-tailed |
| C11 | C18 | 119 | 211 | 417 | 951 | Very Short-tailed |
| C11 | C19 | 117 | 223 | 505 | 1304 | Short-tailed |
| C11 | C20 | 402 | 793 | 1345 | 1979 | Long-tailed |
| C11 | C21 | 106 | 181 | 353 | 852 | Very Short-tailed |
| C11 | C22 | 251 | 560 | 1071 | 1812 | Long-tailed |
| C11 | C23 | 233 | 507 | 1021 | 1809 | Long-tailed |
| C11 | C24 | 109 | 187 | 364 | 859 | Very Short-tailed |
| C11 | C25 | 189 | 435 | 957 | 1783 | Long-tailed |
| C11 | C26 | 106 | 181 | 356 | 884 | Very Short-tailed |
| C12 | C13 | 107 | 183 | 362 | 871 | Very Short-tailed |
| C12 | C14 | 103 | 175 | 336 | 830 | Very Short-tailed |
| C12 | C15 | 108 | 191 | 395 | 1023 | Very Short-tailed |
| C12 | C16 | 100 | 167 | 318 | 772 | Very Short-tailed |
| C12 | C17 | 103 | 175 | 344 | 844 | Very Short-tailed |

| C12 | C18 | 104 | 183 | 369 | 936 | Very Short-tailed |
|-----|-----|-----|-----|------|------|-------------------|
| C12 | C19 | 103 | 177 | 353 | 868 | Very Short-tailed |
| C12 | C20 | 103 | 176 | 342 | 821 | Very Short-tailed |
| C12 | C21 | 101 | 171 | 332 | 798 | Very Short-tailed |
| C12 | C22 | 103 | 174 | 341 | 842 | Very Short-tailed |
| C12 | C23 | 103 | 175 | 342 | 836 | Very Short-tailed |
| C12 | C24 | 107 | 186 | 366 | 873 | Very Short-tailed |
| C12 | C25 | 104 | 177 | 342 | 833 | Very Short-tailed |
| C12 | C26 | 108 | 186 | 366 | 876 | Very Short-tailed |
| C13 | C14 | 302 | 618 | 1157 | 1882 | Long-tailed |
| C13 | C15 | 113 | 195 | 374 | 868 | Very Short-tailed |
| C13 | C16 | 107 | 182 | 359 | 854 | Very Short-tailed |
| C13 | C17 | 367 | 733 | 1279 | 1960 | Long-tailed |
| C13 | C18 | 122 | 220 | 436 | 997 | Very Short-tailed |
| C13 | C19 | 114 | 214 | 482 | 1264 | Short-tailed |
| C13 | C20 | 527 | 951 | 1491 | 2074 | Very Long-tailed |
| C13 | C21 | 107 | 183 | 361 | 869 | Very Short-tailed |
| C13 | C22 | 276 | 626 | 1157 | 1902 | Long-tailed |
| C13 | C23 | 252 | 553 | 1085 | 1854 | Long-tailed |
| C13 | C24 | 111 | 191 | 373 | 880 | Very Short-tailed |
| C13 | C25 | 194 | 460 | 1002 | 1819 | Long-tailed |
| C13 | C26 | 107 | 183 | 357 | 890 | Very Short-tailed |
| C14 | C15 | 264 | 565 | 1120 | 1829 | Long-tailed |
| C14 | C16 | 107 | 185 | 362 | 882 | Very Short-tailed |
| C14 | C18 | 291 | 618 | 1180 | 1883 | Long-tailed |
| C14 | C19 | 188 | 379 | 761 | 1507 | Short-tailed |
| C14 | C21 | 106 | 181 | 349 | 839 | Very Short-tailed |
| C14 | C24 | 131 | 248 | 513 | 1163 | Very Short-tailed |
| C14 | C26 | 127 | 235 | 485 | 1134 | Very Short-tailed |
| C15 | C16 | 111 | 196 | 403 | 1017 | Very Short-tailed |
| C15 | C17 | 185 | 381 | 792 | 1592 | Short-tailed |
| C15 | C18 | 263 | 563 | 1120 | 1825 | Long-tailed |
| C15 | C19 | 142 | 284 | 610 | 1335 | Short-tailed |
| C15 | C20 | 272 | 588 | 1168 | 1885 | Long-tailed |
| C15 | C21 | 109 | 192 | 396 | 1011 | Very Short-tailed |
| C15 | C22 | 192 | 407 | 884 | 1662 | Short-tailed |
| C15 | C23 | 184 | 384 | 789 | 1580 | Short-tailed |
| C15 | C24 | 125 | 235 | 494 | 1181 | Very Short-tailed |
| C15 | C25 | 227 | 494 | 993 | 1746 | Long-tailed |
| C15 | C26 | 129 | 246 | 534 | 1252 | Short-tailed |
| C16 | C17 | 105 | 180 | 357 | 861 | Very Short-tailed |
| C16 | C18 | 108 | 191 | 389 | 966 | Very Short-tailed |
| C16 | C19 | 105 | 182 | 366 | 898 | Very Short-tailed |
| C16 | C20 | 106 | 182 | 360 | 869 | Very Short-tailed |
| C16 | C21 | 101 | 171 | 330 | 792 | Very Short-tailed |
| C16 | C22 | 106 | 183 | 362 | 890 | Very Short-tailed |
| C16 | C23 | 106 | 182 | 359 | 861 | Very Short-tailed |
| C16 | C24 | 107 | 184 | 364 | 881 | Very Short-tailed |
| C16 | C25 | 107 | 184 | 363 | 901 | Very Short-tailed |
| C16 | C26 | 108 | 186 | 369 | 899 | Very Short-tailed |
| C17 | C18 | 215 | 456 | 910 | 1695 | Long-tailed |
| C17 | C19 | 184 | 367 | 737 | 1477 | Short-tailed |
| C17 | C21 | 104 | 176 | 345 | 842 | Very Short-tailed |
| C17 | C24 | 122 | 223 | 458 | 1066 | Very Short-tailed |
| C17 | C26 | 118 | 213 | 441 | 1022 | Very Short-tailed |
| C18 | C19 | 166 | 348 | 742 | 1558 | Short-tailed |
| C18 | C20 | 335 | 712 | 1292 | 2002 | Long-tailed |
| C18 | C21 | 106 | 185 | 374 | 936 | Very Short-tailed |
| C18 | C22 | 211 | 454 | 958 | 1722 | Long-tailed |
| C18 | C23 | 217 | 460 | 918 | 1684 | Long-tailed |
| C18 | C24 | 126 | 238 | 497 | 1204 | Very Short-tailed |
| C18 | C25 | 264 | 564 | 1119 | 1860 | Long-tailed |
| C18 | C26 | 130 | 251 | 528 | 1253 | Short-tailed |
| C19 | C20 | 287 | 573 | 1076 | 1773 | Long-tailed |
| C19 | C21 | 104 | 180 | 358 | 870 | Very Short-tailed |
| C19 | C22 | 206 | 453 | 888 | 1648 | Long-tailed |
| C19 | C23 | 193 | 391 | 779 | 1526 | Short-tailed |
| C19 | C24 | 120 | 225 | 473 | 1125 | Very Short-tailed |
| C19 | C25 | 166 | 344 | 724 | 1575 | Short-tailed |
| C19 | C26 | 116 | 205 | 417 | 1004 | Very Short-tailed |
| C20 | C21 | 104 | 179 | 348 | 830 | Very Short-tailed |
| C20 | C24 | 130 | 247 | 516 | 1171 | Very Short-tailed |
| C20 | C26 | 130 | 241 | 498 | 1140 | Very Short-tailed |
| C21 | C22 | 104 | 177 | 345 | 848 | Very Short-tailed |
| C21 | C23 | 104 | 177 | 346 | 846 | Very Short-tailed |
| C21 | C24 | 108 | 187 | 368 | 864 | Very Short-tailed |
| C21 | C25 | 105 | 180 | 349 | 845 | Very Short-tailed |
| C21 | C26 | 108 | 186 | 366 | 875 | Very Short-tailed |
| C22 | C24 | 127 | 237 | 504 | 1170 | Very Short-tailed |
| C22 | C26 | 124 | 230 | 482 | 1124 | Very Short-tailed |
| C23 | C24 | 123 | 226 | 467 | 1079 | Very Short-tailed |
| C23 | C26 | 122 | 221 | 445 | 1039 | Very Short-tailed |
| C24 | C25 | 133 | 255 | 537 | 1257 | Short-tailed |
| C24 | C26 | 134 | 256 | 540 | 1255 | Short-tailed |
| C25 | C26 | 133 | 250 | 523 | 1197 | Short-tailed |
| C2 | - | 934 | 1443 | 1846 | 2283 | - |

| | | | | | |
|---|---|---|---|---|---|
| C5 | - | 140 | 280 | 630 | 1550 | - |
| C6 | - | 573 | 1362 | 2079 | 2079 | - |
| C14 | - | 1657 | 1657 | 1657 | 1657 | - |
| C17 | - | 1292 | 1292 | 1292 | 1292 | - |
| C22 | - | 117 | 180 | 496 | 496 | - |
| C23 | - | 62 | 62 | 62 | 62 | - |
| C25 | - | 190 | 408 | 591 | 730 | - |
| C2 | C9 | 544 | 993 | 1642 | 2277 | - |
| C2 | C20 | 801 | 1200 | 1713 | 2249 | - |
| C5 | C9 | 143 | 303 | 726 | 1509 | - |
| C5 | C20 | 168 | 346 | 793 | 1708 | - |
| C5 | C22 | 154 | 327 | 712 | 1467 | - |
| C6 | C9 | 459 | 771 | 1236 | 1852 | - |
| C6 | C14 | 259 | 608 | 1545 | 2127 | - |
| C6 | C17 | 209 | 353 | 620 | 864 | - |
| C6 | C20 | 336 | 747 | 1060 | 1738 | - |
| C6 | C22 | 413 | 573 | 1362 | 1869 | - |
| C6 | C23 | 358 | 717 | 1006 | 1902 | - |
| C6 | C25 | 355 | 817 | 1475 | 2060 | - |
| C9 | C14 | 613 | 1046 | 1657 | 2380 | - |
| C9 | C17 | 660 | 1513 | 1877 | 2286 | - |
| C9 | C20 | 773 | 1902 | 1902 | 1902 | - |
| C9 | C22 | 457 | 1115 | 1169 | 1302 | - |
| C9 | C23 | 294 | 917 | 1526 | 2323 | - |
| C9 | C25 | 302 | 641 | 1361 | 2223 | - |
| C14 | C17 | 822 | 1320 | 1792 | 2134 | - |
| C14 | C20 | 530 | 737 | 1657 | 1988 | - |
| C14 | C22 | 296 | 958 | 1527 | 2076 | - |
| C14 | C23 | 574 | 944 | 1781 | 2305 | - |
| C14 | C25 | 299 | 700 | 1304 | 1909 | - |
| C17 | C20 | 367 | 367 | 513 | 513 | - |
| C17 | C22 | 369 | 580 | 830 | 1655 | - |
| C17 | C23 | 247 | 619 | 912 | 1733 | - |
| C17 | C25 | 471 | 885 | 1691 | 2262 | - |
| C20 | C22 | 812 | 1482 | 2174 | 2180 | - |
| C20 | C23 | 1016 | 1386 | 1700 | 1700 | - |
| C20 | C25 | 297 | 834 | 1605 | 2243 | - |
| C22 | C23 | 476 | 768 | 1291 | 1899 | - |
| C22 | C25 | 310 | 536 | 1147 | 1903 | - |
| C23 | C25 | 348 | 740 | 1327 | 2105 | - |

## A.10 Average Log Likelihood of Label Histogram Distributions for all groupings

This table contains the Average Log Likelihood of Label Proportion distributions for all $349$ groupings (removing groupings which were left with no bags after clipping). We perform K-Means of $308$ of these groupings that have more than $500$ bags after clipping. The cluster assigned to each bag is also listed.

Table 12: Average Log Likelihood of Label Distribution of all grouping and it's clusters

| Col1 | Col2 | Standard deviation of label proportions | Label bias of the grouping | Average Log likelihood of label histogram distribution | Clusters assigned on label histogram diatribution |
|---|---|---|---|---|---|
| C1 | - | 0.05 | 0.26 | -3.31 | High |
| C3 | - | 0.17 | 0.26 | -17.26 | Low |
| C4 | - | 0.16 | 0.26 | -18.24 | Low |
| C7 | - | 0.11 | 0.22 | -25.85 | Very Low |
| C8 | - | 0.05 | 0.26 | -3.33 | High |
| C10 | - | 0.1 | 0.25 | -11.68 | Medium |
| C11 | - | 0.1 | 0.21 | -29.13 | Very Low |
| C12 | - | 0.17 | 0.26 | -17.29 | Low |
| C13 | - | 0.09 | 0.21 | -32.14 | Very Low |
| C15 | - | 0.16 | 0.27 | -34.69 | Very Low |
| C16 | - | 0.16 | 0.26 | -17.37 | Low |
| C18 | - | 0.15 | 0.29 | -35.47 | Very Low |
| C19 | - | 0.07 | 0.26 | -10.9 | Medium |
| C21 | - | 0.16 | 0.26 | -17.33 | Low |
| C24 | - | 0.14 | 0.26 | -19.06 | Low |
| C26 | - | 0.14 | 0.25 | -17.84 | Low |
| C1 | C2 | 0.1 | 0.26 | -11.58 | Medium |
| C1 | C3 | 0.16 | 0.26 | -15.35 | Medium |
| C1 | C4 | 0.15 | 0.26 | -14.82 | Medium |
| C1 | C5 | 0.05 | 0.26 | -3.29 | High |
| C1 | C6 | 0.05 | 0.26 | -3.86 | High |
| C1 | C7 | 0.12 | 0.24 | -14.11 | Medium |
| C1 | C8 | 0.05 | 0.26 | -3.26 | High |
| C1 | C9 | 0.08 | 0.24 | -6.41 | High |
| C1 | C10 | 0.1 | 0.26 | -9.17 | High |
| C1 | C11 | 0.11 | 0.24 | -13.62 | Medium |
| C1 | C12 | 0.16 | 0.26 | -15.3 | Medium |

| C1 | C13 | 0.11 | 0.24 | -12.79 | Medium |
|---|---|---|---|---|---|
| C1 | C14 | 0.1 | 0.26 | -10.68 | Medium |
| C1 | C15 | 0.14 | 0.27 | -17.63 | Low |
| C1 | C16 | 0.15 | 0.26 | -14.98 | Medium |
| C1 | C17 | 0.09 | 0.23 | -9.84 | Medium |
| C1 | C18 | 0.14 | 0.27 | -16.43 | Low |
| C1 | C19 | 0.08 | 0.26 | -6.54 | High |
| C1 | C20 | 0.05 | 0.25 | -3.74 | High |
| C1 | C21 | 0.16 | 0.26 | -15.25 | Medium |
| C1 | C22 | 0.05 | 0.26 | -3.47 | High |
| C1 | C23 | 0.08 | 0.25 | -8.35 | High |
| C1 | C24 | 0.14 | 0.26 | -15.09 | Medium |
| C1 | C25 | 0.06 | 0.27 | -4.95 | High |
| C1 | C26 | 0.14 | 0.25 | -14.47 | Medium |
| C2 | C3 | 0.17 | 0.26 | -18.51 | Low |
| C2 | C4 | 0.16 | 0.25 | -18.73 | Low |
| C2 | C5 | 0.1 | 0.26 | -13.44 | Medium |
| C2 | C6 | 0.12 | 0.26 | -25.47 | Very Low |
| C2 | C7 | 0.14 | 0.24 | -17.67 | Low |
| C2 | C8 | 0.1 | 0.26 | -12.51 | Medium |
| C2 | C10 | 0.13 | 0.26 | -12.53 | Medium |
| C2 | C11 | 0.14 | 0.24 | -19.45 | Low |
| C2 | C12 | 0.17 | 0.26 | -18.55 | Low |
| C2 | C13 | 0.14 | 0.24 | -19.35 | Low |
| C2 | C14 | 0.13 | 0.23 | -38.88 | Very Low |
| C2 | C15 | 0.16 | 0.27 | -34.68 | Very Low |
| C2 | C16 | 0.17 | 0.26 | -18.51 | Low |
| C2 | C17 | 0.12 | 0.22 | -30.65 | Very Low |
| C2 | C18 | 0.15 | 0.29 | -35.45 | Very Low |
| C2 | C19 | 0.1 | 0.26 | -10.64 | Medium |
| C2 | C21 | 0.17 | 0.26 | -18.49 | Low |
| C2 | C22 | 0.12 | 0.27 | -20.9 | Low |
| C2 | C23 | 0.13 | 0.23 | -31.79 | Very Low |
| C2 | C24 | 0.15 | 0.25 | -19.33 | Low |
| C2 | C25 | 0.11 | 0.27 | -16.42 | Low |
| C2 | C26 | 0.15 | 0.24 | -17.86 | Low |
| C3 | C4 | 0.16 | 0.26 | -16.04 | Low |
| C3 | C5 | 0.16 | 0.26 | -16.29 | Low |
| C3 | C6 | 0.17 | 0.26 | -17 | Low |
| C3 | C7 | 0.18 | 0.25 | -17.95 | Low |
| C3 | C8 | 0.16 | 0.26 | -15.91 | Medium |
| C3 | C9 | 0.17 | 0.26 | -18.56 | Low |
| C3 | C10 | 0.16 | 0.26 | -14.44 | Medium |
| C3 | C11 | 0.18 | 0.25 | -18.57 | Low |
| C3 | C12 | 0.17 | 0.26 | -17.24 | Low |
| C3 | C13 | 0.18 | 0.24 | -18.5 | Low |
| C3 | C14 | 0.18 | 0.26 | -19.68 | Low |
| C3 | C15 | 0.18 | 0.26 | -21.58 | Low |
| C3 | C16 | 0.16 | 0.26 | -16.08 | Low |
| C3 | C17 | 0.18 | 0.26 | -19.53 | Low |
| C3 | C18 | 0.17 | 0.26 | -19.44 | Low |
| C3 | C19 | 0.16 | 0.26 | -16.79 | Low |
| C3 | C20 | 0.17 | 0.26 | -18.1 | Low |
| C3 | C21 | 0.17 | 0.26 | -16.86 | Low |
| C3 | C22 | 0.17 | 0.26 | -17.05 | Low |
| C3 | C23 | 0.18 | 0.26 | -19.2 | Low |
| C3 | C24 | 0.16 | 0.26 | -16.42 | Low |
| C3 | C25 | 0.17 | 0.26 | -17.98 | Low |
| C3 | C26 | 0.16 | 0.26 | -17.63 | Low |
| C4 | C5 | 0.15 | 0.26 | -15.94 | Medium |
| C4 | C6 | 0.16 | 0.26 | -16.48 | Low |
| C4 | C7 | 0.18 | 0.26 | -17.39 | Low |
| C4 | C8 | 0.15 | 0.26 | -15.44 | Medium |
| C4 | C9 | 0.16 | 0.25 | -19.17 | Low |
| C4 | C10 | 0.16 | 0.27 | -14.9 | Medium |
| C4 | C11 | 0.17 | 0.26 | -17.92 | Low |
| C4 | C12 | 0.16 | 0.26 | -15.98 | Medium |
| C4 | C13 | 0.17 | 0.26 | -17.93 | Low |
| C4 | C14 | 0.16 | 0.26 | -19.6 | Low |
| C4 | C15 | 0.17 | 0.26 | -20.63 | Low |
| C4 | C16 | 0.16 | 0.26 | -16.51 | Low |
| C4 | C17 | 0.16 | 0.26 | -18.79 | Low |
| C4 | C18 | 0.16 | 0.26 | -19.2 | Low |
| C4 | C19 | 0.15 | 0.26 | -17 | Low |
| C4 | C20 | 0.16 | 0.26 | -18.48 | Low |
| C4 | C21 | 0.16 | 0.26 | -15.96 | Medium |
| C4 | C22 | 0.16 | 0.26 | -17.34 | Low |
| C4 | C23 | 0.17 | 0.26 | -19.01 | Low |
| C4 | C24 | 0.15 | 0.26 | -17.38 | Low |
| C4 | C25 | 0.16 | 0.26 | -18.53 | Low |
| C4 | C26 | 0.16 | 0.26 | -18.41 | Low |
| C5 | C6 | 0.05 | 0.26 | -4.13 | High |
| C5 | C7 | 0.12 | 0.23 | -15.47 | Medium |
| C5 | C8 | 0.05 | 0.26 | -3.29 | High |
| C5 | C10 | 0.1 | 0.26 | -9.8 | Medium |
| C5 | C11 | 0.11 | 0.23 | -15.13 | Medium |

| C5 | C12 | 0.16 | 0.26 | -16.24 | Low |
|----|-----|------|------|--------|-----|
| C5 | C13 | 0.11 | 0.23 | -13.94 | Medium |
| C5 | C14 | 0.11 | 0.26 | -13.34 | Medium |
| C5 | C15 | 0.15 | 0.27 | -20.47 | Low |
| C5 | C16 | 0.16 | 0.26 | -16.02 | Low |
| C5 | C17 | 0.09 | 0.23 | -11.8 | Medium |
| C5 | C18 | 0.14 | 0.28 | -19.16 | Low |
| C5 | C19 | 0.07 | 0.26 | -7.35 | High |
| C5 | C21 | 0.16 | 0.26 | -16.28 | Low |
| C5 | C23 | 0.08 | 0.24 | -9.94 | Medium |
| C5 | C24 | 0.14 | 0.26 | -16.28 | Low |
| C5 | C25 | 0.06 | 0.27 | -5.53 | High |
| C5 | C26 | 0.14 | 0.25 | -15.58 | Medium |
| C6 | C7 | 0.12 | 0.23 | -18.8 | Low |
| C6 | C8 | 0.05 | 0.26 | -3.94 | High |
| C6 | C10 | 0.1 | 0.26 | -10.84 | Medium |
| C6 | C11 | 0.11 | 0.23 | -21.32 | Low |
| C6 | C12 | 0.16 | 0.26 | -16.94 | Low |
| C6 | C13 | 0.1 | 0.23 | -20.68 | Low |
| C6 | C15 | 0.16 | 0.27 | -26.42 | Very Low |
| C6 | C16 | 0.16 | 0.26 | -16.72 | Low |
| C6 | C18 | 0.16 | 0.29 | -26.28 | Very Low |
| C6 | C19 | 0.08 | 0.26 | -8.89 | High |
| C6 | C21 | 0.16 | 0.26 | -16.91 | Low |
| C6 | C24 | 0.15 | 0.26 | -17.56 | Low |
| C6 | C26 | 0.15 | 0.25 | -16.74 | Low |
| C7 | C8 | 0.12 | 0.24 | -14.95 | Medium |
| C7 | C9 | 0.11 | 0.22 | -25.84 | Very Low |
| C7 | C10 | 0.11 | 0.24 | -13.29 | Medium |
| C7 | C11 | 0.11 | 0.22 | -25.75 | Very Low |
| C7 | C12 | 0.17 | 0.25 | -17.65 | Low |
| C7 | C13 | 0.11 | 0.22 | -25.85 | Very Low |
| C7 | C14 | 0.14 | 0.23 | -21.55 | Low |
| C7 | C15 | 0.17 | 0.25 | -17.15 | Low |
| C7 | C16 | 0.17 | 0.25 | -17.37 | Low |
| C7 | C17 | 0.11 | 0.21 | -20.27 | Low |
| C7 | C18 | 0.16 | 0.24 | -17.44 | Low |
| C7 | C19 | 0.14 | 0.24 | -16.64 | Low |
| C7 | C20 | 0.12 | 0.22 | -20.59 | Low |
| C7 | C21 | 0.18 | 0.25 | -17.68 | Low |
| C7 | C22 | 0.12 | 0.24 | -19.64 | Low |
| C7 | C23 | 0.12 | 0.23 | -20 | Low |
| C7 | C24 | 0.17 | 0.26 | -16.83 | Low |
| C7 | C25 | 0.12 | 0.23 | -19.57 | Low |
| C7 | C26 | 0.17 | 0.25 | -17.49 | Low |
| C8 | C9 | 0.08 | 0.23 | -7.1 | High |
| C8 | C10 | 0.1 | 0.26 | -9.48 | High |
| C8 | C11 | 0.11 | 0.24 | -14.45 | Medium |
| C8 | C12 | 0.16 | 0.26 | -15.84 | Medium |
| C8 | C13 | 0.11 | 0.24 | -13.45 | Medium |
| C8 | C14 | 0.1 | 0.26 | -12.16 | Medium |
| C8 | C15 | 0.14 | 0.27 | -19.11 | Low |
| C8 | C16 | 0.16 | 0.26 | -15.6 | Medium |
| C8 | C17 | 0.09 | 0.23 | -10.55 | Medium |
| C8 | C18 | 0.14 | 0.28 | -17.86 | Low |
| C8 | C19 | 0.08 | 0.26 | -6.9 | High |
| C8 | C20 | 0.05 | 0.25 | -3.84 | High |
| C8 | C21 | 0.16 | 0.26 | -15.87 | Medium |
| C8 | C22 | 0.05 | 0.26 | -3.54 | High |
| C8 | C23 | 0.08 | 0.24 | -9 | High |
| C8 | C24 | 0.14 | 0.26 | -15.74 | Medium |
| C8 | C25 | 0.06 | 0.27 | -5.27 | High |
| C8 | C26 | 0.14 | 0.25 | -15.18 | Medium |
| C9 | C10 | 0.1 | 0.24 | -12.92 | Medium |
| C9 | C11 | 0.1 | 0.2 | -30.29 | Very Low |
| C9 | C12 | 0.17 | 0.26 | -18.55 | Low |
| C9 | C13 | 0.1 | 0.19 | -33.86 | Very Low |
| C9 | C15 | 0.16 | 0.25 | -34.98 | Very Low |
| C9 | C16 | 0.17 | 0.26 | -18.45 | Low |
| C9 | C18 | 0.16 | 0.26 | -35.17 | Very Low |
| C9 | C19 | 0.1 | 0.23 | -15.93 | Medium |
| C9 | C21 | 0.17 | 0.26 | -18.52 | Low |
| C9 | C24 | 0.15 | 0.25 | -20.43 | Low |
| C9 | C26 | 0.14 | 0.24 | -18.93 | Low |
| C10 | C11 | 0.1 | 0.24 | -13.61 | Medium |
| C10 | C12 | 0.16 | 0.26 | -14.31 | Medium |
| C10 | C13 | 0.1 | 0.24 | -12.8 | Medium |
| C10 | C14 | 0.12 | 0.26 | -13.33 | Medium |
| C10 | C15 | 0.15 | 0.26 | -14.08 | Medium |
| C10 | C16 | 0.16 | 0.26 | -14.35 | Medium |
| C10 | C17 | 0.1 | 0.25 | -11.73 | Medium |
| C10 | C18 | 0.15 | 0.26 | -13.55 | Medium |
| C10 | C19 | 0.12 | 0.26 | -11.26 | Medium |
| C10 | C20 | 0.1 | 0.26 | -11.43 | Medium |
| C10 | C21 | 0.16 | 0.26 | -14.4 | Medium |
| C10 | C22 | 0.1 | 0.26 | -11.19 | Medium |

| C10 | C23 | 0.11 | 0.26 | -12.46 | Medium |
|-----|-----|------|------|--------|--------|
| C10 | C24 | 0.15 | 0.27 | -13.68 | Medium |
| C10 | C25 | 0.11 | 0.26 | -11.28 | Medium |
| C10 | C26 | 0.15 | 0.26 | -13.96 | Medium |
| C11 | C12 | 0.18 | 0.25 | -18.45 | Low |
| C11 | C13 | 0.1 | 0.21 | -29.1 | Very Low |
| C11 | C14 | 0.14 | 0.23 | -25.74 | Very Low |
| C11 | C15 | 0.16 | 0.24 | -17.9 | Low |
| C11 | C16 | 0.17 | 0.25 | -18.17 | Low |
| C11 | C17 | 0.11 | 0.21 | -20.9 | Low |
| C11 | C18 | 0.16 | 0.24 | -18.61 | Low |
| C11 | C19 | 0.13 | 0.25 | -14.39 | Medium |
| C11 | C20 | 0.11 | 0.22 | -23.87 | Low |
| C11 | C21 | 0.18 | 0.25 | -18.6 | Low |
| C11 | C22 | 0.12 | 0.23 | -20.17 | Low |
| C11 | C23 | 0.12 | 0.23 | -21 | Low |
| C11 | C24 | 0.17 | 0.25 | -17.53 | Low |
| C11 | C25 | 0.12 | 0.23 | -18.45 | Low |
| C11 | C26 | 0.17 | 0.25 | -17.67 | Low |
| C12 | C13 | 0.17 | 0.25 | -18.42 | Low |
| C12 | C14 | 0.18 | 0.26 | -19.52 | Low |
| C12 | C15 | 0.18 | 0.26 | -21.37 | Low |
| C12 | C16 | 0.16 | 0.26 | -16.1 | Low |
| C12 | C17 | 0.17 | 0.26 | -19.4 | Low |
| C12 | C18 | 0.17 | 0.26 | -19.37 | Low |
| C12 | C19 | 0.16 | 0.26 | -16.82 | Low |
| C12 | C20 | 0.17 | 0.26 | -18.09 | Low |
| C12 | C21 | 0.16 | 0.26 | -16.99 | Low |
| C12 | C22 | 0.16 | 0.26 | -17.03 | Low |
| C12 | C23 | 0.18 | 0.26 | -19.21 | Low |
| C12 | C24 | 0.16 | 0.26 | -16.4 | Low |
| C12 | C25 | 0.17 | 0.26 | -18.02 | Low |
| C12 | C26 | 0.16 | 0.26 | -17.62 | Low |
| C13 | C14 | 0.13 | 0.23 | -26.13 | Very Low |
| C13 | C15 | 0.16 | 0.24 | -18.38 | Low |
| C13 | C16 | 0.17 | 0.25 | -18.16 | Low |
| C13 | C17 | 0.1 | 0.21 | -20.48 | Low |
| C13 | C18 | 0.16 | 0.24 | -18.75 | Low |
| C13 | C19 | 0.13 | 0.25 | -13.32 | Medium |
| C13 | C20 | 0.1 | 0.22 | -24.19 | Low |
| C13 | C21 | 0.17 | 0.25 | -18.56 | Low |
| C13 | C22 | 0.11 | 0.23 | -19.33 | Low |
| C13 | C23 | 0.12 | 0.23 | -20.39 | Low |
| C13 | C24 | 0.16 | 0.25 | -17.45 | Low |
| C13 | C25 | 0.12 | 0.23 | -17.36 | Low |
| C13 | C26 | 0.17 | 0.25 | -17.96 | Low |
| C14 | C15 | 0.16 | 0.27 | -34.64 | Very Low |
| C14 | C16 | 0.17 | 0.26 | -19.62 | Low |
| C14 | C18 | 0.15 | 0.27 | -34.77 | Very Low |
| C14 | C19 | 0.11 | 0.25 | -13.26 | Medium |
| C14 | C21 | 0.18 | 0.26 | -19.73 | Low |
| C14 | C24 | 0.16 | 0.26 | -21 | Low |
| C14 | C26 | 0.15 | 0.25 | -19.3 | Low |
| C15 | C16 | 0.17 | 0.26 | -21.13 | Low |
| C15 | C17 | 0.16 | 0.25 | -29.03 | Very Low |
| C15 | C18 | 0.16 | 0.27 | -34.66 | Very Low |
| C15 | C19 | 0.13 | 0.26 | -17.46 | Low |
| C15 | C20 | 0.16 | 0.27 | -34.75 | Very Low |
| C15 | C21 | 0.18 | 0.26 | -21.28 | Low |
| C15 | C22 | 0.16 | 0.28 | -26.58 | Very Low |
| C15 | C23 | 0.16 | 0.26 | -30.28 | Very Low |
| C15 | C24 | 0.16 | 0.25 | -21.78 | Low |
| C15 | C25 | 0.15 | 0.27 | -28.49 | Very Low |
| C15 | C26 | 0.16 | 0.25 | -21.96 | Low |
| C16 | C17 | 0.17 | 0.26 | -18.97 | Low |
| C16 | C18 | 0.17 | 0.26 | -19.3 | Low |
| C16 | C19 | 0.16 | 0.26 | -16.83 | Low |
| C16 | C20 | 0.17 | 0.26 | -18.13 | Low |
| C16 | C21 | 0.16 | 0.26 | -16.43 | Low |
| C16 | C22 | 0.16 | 0.26 | -17.16 | Low |
| C16 | C23 | 0.17 | 0.26 | -19.05 | Low |
| C16 | C24 | 0.16 | 0.26 | -16.29 | Low |
| C16 | C25 | 0.16 | 0.26 | -17.99 | Low |
| C16 | C26 | 0.16 | 0.26 | -17.68 | Low |
| C17 | C18 | 0.16 | 0.26 | -30.52 | Very Low |
| C17 | C19 | 0.1 | 0.24 | -13 | Medium |
| C17 | C21 | 0.17 | 0.26 | -19.25 | Low |
| C17 | C24 | 0.16 | 0.25 | -20.31 | Low |
| C17 | C26 | 0.15 | 0.24 | -18.47 | Low |
| C18 | C19 | 0.12 | 0.27 | -15.5 | Medium |
| C18 | C20 | 0.15 | 0.28 | -35.7 | Very Low |
| C18 | C21 | 0.17 | 0.26 | -19.29 | Low |
| C18 | C22 | 0.15 | 0.29 | -25.63 | Very Low |
| C18 | C23 | 0.16 | 0.26 | -32.27 | Very Low |
| C18 | C24 | 0.15 | 0.25 | -20.08 | Low |
| C18 | C25 | 0.14 | 0.28 | -28.31 | Very Low |

| C18 | C26 | 0.15 | 0.25 | -20 | Low |
|-----|-----|------|------|-----|-----|
| C19 | C20 | 0.08 | 0.26 | -10.84 | Medium |
| C19 | C21 | 0.16 | 0.26 | -16.92 | Low |
| C19 | C22 | 0.08 | 0.26 | -8.94 | High |
| C19 | C23 | 0.1 | 0.25 | -12.29 | Medium |
| C19 | C24 | 0.14 | 0.25 | -16.5 | Low |
| C19 | C25 | 0.09 | 0.26 | -10.23 | Medium |
| C19 | C26 | 0.14 | 0.25 | -14.99 | Medium |
| C20 | C21 | 0.17 | 0.26 | -18.19 | Low |
| C20 | C24 | 0.15 | 0.25 | -19.38 | Low |
| C20 | C26 | 0.15 | 0.25 | -18.09 | Low |
| C21 | C22 | 0.16 | 0.26 | -17.08 | Low |
| C21 | C23 | 0.18 | 0.26 | -19.2 | Low |
| C21 | C24 | 0.16 | 0.26 | -16.34 | Low |
| C21 | C25 | 0.17 | 0.26 | -17.94 | Low |
| C21 | C26 | 0.16 | 0.26 | -17.63 | Low |
| C22 | C24 | 0.15 | 0.26 | -18.31 | Low |
| C22 | C26 | 0.14 | 0.25 | -17.29 | Low |
| C23 | C24 | 0.16 | 0.26 | -20.12 | Low |
| C23 | C26 | 0.15 | 0.24 | -18.82 | Low |
| C24 | C25 | 0.15 | 0.25 | -19.27 | Low |
| C24 | C26 | 0.15 | 0.25 | -20.44 | Low |
| C25 | C26 | 0.14 | 0.24 | -18.44 | Low |
| C2 | - | 0.13 | 0.28 | -47.05 | - |
| C5 | - | 0.05 | 0.26 | -3.41 | - |
| C6 | - | 0.09 | 0.25 | -12.4 | - |
| C14 | - | 0.12 | 0.29 | -33.6 | - |
| C17 | - | 0 | 0.06 | -3.09 | - |
| C22 | - | 0.05 | 0.26 | -3.24 | - |
| C23 | - | 0 | 0.11 | -1.84 | - |
| C25 | - | 0.08 | 0.26 | -5.81 | - |
| C2 | C9 | 0.12 | 0.21 | -36.84 | - |
| C2 | C20 | 0.13 | 0.27 | -42.36 | - |
| C5 | C9 | 0.08 | 0.22 | -8.52 | - |
| C5 | C20 | 0.04 | 0.25 | -3.93 | - |
| C5 | C22 | 0.05 | 0.27 | -3.67 | - |
| C6 | C9 | 0.09 | 0.19 | -16.45 | - |
| C6 | C14 | 0.13 | 0.23 | -32.2 | - |
| C6 | C17 | 0.09 | 0.2 | -11.09 | - |
| C6 | C20 | 0.07 | 0.23 | -5.86 | - |
| C6 | C22 | 0.15 | 0.27 | -9.38 | - |
| C6 | C23 | 0.11 | 0.2 | -21.74 | - |
| C6 | C25 | 0.08 | 0.28 | -9.46 | - |
| C9 | C14 | 0.11 | 0.22 | -36.01 | - |
| C9 | C17 | 0.06 | 0.11 | -16.34 | - |
| C9 | C20 | 0.03 | 0.11 | -10.53 | - |
| C9 | C22 | 0.08 | 0.18 | -7.53 | - |
| C9 | C23 | 0.04 | 0.12 | -10.5 | - |
| C9 | C25 | 0.1 | 0.16 | -21.25 | - |
| C14 | C17 | 0.14 | 0.22 | -65.29 | - |
| C14 | C20 | 0.13 | 0.2 | -38.31 | - |
| C14 | C22 | 0.15 | 0.26 | -37.84 | - |
| C14 | C23 | 0.16 | 0.24 | -66.83 | - |
| C14 | C25 | 0.13 | 0.26 | -26.84 | - |
| C17 | C20 | 0.02 | 0.06 | -3.74 | - |
| C17 | C22 | 0.11 | 0.25 | -15.52 | - |
| C17 | C23 | 0.06 | 0.09 | -10.64 | - |
| C17 | C25 | 0.11 | 0.21 | -25.85 | - |
| C20 | C22 | 0.05 | 0.29 | -4.91 | - |
| C20 | C23 | 0.16 | 0.11 | -78.7 | - |
| C20 | C25 | 0.08 | 0.25 | -11.92 | - |
| C22 | C23 | 0.11 | 0.26 | -21.53 | - |
| C22 | C25 | 0.08 | 0.31 | -9.01 | - |
| C23 | C25 | 0.12 | 0.24 | -27.2 | - |

## A.11 BAG SEPARATION STATISTICS FOR ALL THE GROUPINGS

This table contains MeanInterBagSep, MeanIntraBagSep and their ratio for all 349 clipped groupings (removing groupings which were left with no bags after clipping). We perform K-Means of 308 of these groupings that have more than 500 bags after clipping. The cluster assigned to each bag is also listed.

Table 13: Bag Separation Statistics and their clusters on all groupings

| Col1 | Col2 | MeanInterBagSep | MeanIntraBagSep | InterIntraRatio | Clusters assigned based on InterIntraRatio distribution |
|------|------|-----------------|-----------------|-----------------|--------------------------------------------------------|
| C1 | - | 0.83 | 0.82 | 1.02 | Less-separated |
| C3 | - | 0.81 | 0.7 | 1.16 | Medium-separated |
| C4 | - | 0.81 | 0.7 | 1.16 | Medium-separated |
| C7 | - | 0.81 | 0.61 | 1.33 | Well-separated |
| C8 | - | 0.83 | 0.81 | 1.02 | Less-separated |
| C10 | - | 0.82 | 0.7 | 1.18 | Medium-separated |

| C11 | - | 0.75 | 0.63 | 1.2 | Medium-separated |
|-----|-----|------|------|------|------------------|
| C12 | - | 0.81 | 0.7 | 1.16 | Medium-separated |
| C13 | - | 0.72 | 0.65 | 1.1 | Less-separated |
| C15 | - | 0.8 | 0.7 | 1.15 | Medium-separated |
| C16 | - | 0.81 | 0.7 | 1.16 | Medium-separated |
| C18 | - | 0.79 | 0.71 | 1.12 | Medium-separated |
| C19 | - | 0.78 | 0.76 | 1.03 | Less-separated |
| C21 | - | 0.81 | 0.7 | 1.16 | Medium-separated |
| C24 | - | 0.82 | 0.72 | 1.14 | Medium-separated |
| C26 | - | 0.79 | 0.7 | 1.12 | Medium-separated |
| C1 | C2 | 0.81 | 0.74 | 1.1 | Less-separated |
| C1 | C3 | 0.83 | 0.72 | 1.15 | Medium-separated |
| C1 | C4 | 0.82 | 0.72 | 1.14 | Medium-separated |
| C1 | C5 | 0.83 | 0.81 | 1.02 | Less-separated |
| C1 | C6 | 0.82 | 0.79 | 1.03 | Less-separated |
| C1 | C7 | 0.81 | 0.6 | 1.34 | Well-separated |
| C1 | C8 | 0.83 | 0.81 | 1.02 | Less-separated |
| C1 | C9 | 0.9 | 0.8 | 1.12 | Medium-separated |
| C1 | C10 | 0.82 | 0.73 | 1.14 | Medium-separated |
| C1 | C11 | 0.79 | 0.68 | 1.17 | Medium-separated |
| C1 | C12 | 0.83 | 0.72 | 1.15 | Medium-separated |
| C1 | C13 | 0.79 | 0.7 | 1.13 | Medium-separated |
| C1 | C14 | 0.82 | 0.78 | 1.05 | Less-separated |
| C1 | C15 | 0.82 | 0.72 | 1.14 | Medium-separated |
| C1 | C16 | 0.82 | 0.72 | 1.15 | Medium-separated |
| C1 | C17 | 0.86 | 0.73 | 1.17 | Medium-separated |
| C1 | C18 | 0.81 | 0.73 | 1.12 | Medium-separated |
| C1 | C19 | 0.8 | 0.77 | 1.04 | Less-separated |
| C1 | C20 | 0.83 | 0.81 | 1.03 | Less-separated |
| C1 | C21 | 0.83 | 0.72 | 1.15 | Medium-separated |
| C1 | C22 | 0.83 | 0.81 | 1.02 | Less-separated |
| C1 | C23 | 0.84 | 0.8 | 1.05 | Less-separated |
| C1 | C24 | 0.83 | 0.73 | 1.13 | Medium-separated |
| C1 | C25 | 0.83 | 0.8 | 1.04 | Less-separated |
| C1 | C26 | 0.81 | 0.72 | 1.12 | Medium-separated |
| C2 | C3 | 0.82 | 0.69 | 1.19 | Medium-separated |
| C2 | C4 | 0.82 | 0.69 | 1.18 | Medium-separated |
| C2 | C5 | 0.81 | 0.73 | 1.1 | Less-separated |
| C2 | C6 | 0.74 | 0.66 | 1.13 | Medium-separated |
| C2 | C7 | 0.78 | 0.55 | 1.44 | Far-separated |
| C2 | C8 | 0.81 | 0.74 | 1.1 | Less-separated |
| C2 | C10 | 0.81 | 0.67 | 1.21 | Medium-separated |
| C2 | C11 | 0.76 | 0.6 | 1.26 | Well-separated |
| C2 | C12 | 0.82 | 0.69 | 1.19 | Medium-separated |
| C2 | C13 | 0.75 | 0.62 | 1.22 | Medium-separated |
| C2 | C14 | 0.76 | 0.68 | 1.13 | Medium-separated |
| C2 | C15 | 0.8 | 0.7 | 1.15 | Medium-separated |
| C2 | C16 | 0.82 | 0.69 | 1.18 | Medium-separated |
| C2 | C17 | 0.81 | 0.6 | 1.34 | Well-separated |
| C2 | C18 | 0.79 | 0.71 | 1.12 | Medium-separated |
| C2 | C19 | 0.78 | 0.72 | 1.09 | Less-separated |
| C2 | C21 | 0.82 | 0.69 | 1.19 | Medium-separated |
| C2 | C22 | 0.77 | 0.69 | 1.12 | Medium-separated |
| C2 | C23 | 0.79 | 0.66 | 1.2 | Medium-separated |
| C2 | C24 | 0.83 | 0.71 | 1.17 | Medium-separated |
| C2 | C25 | 0.85 | 0.78 | 1.09 | Less-separated |
| C2 | C26 | 0.8 | 0.7 | 1.14 | Medium-separated |
| C3 | C4 | 0.8 | 0.69 | 1.16 | Medium-separated |
| C3 | C5 | 0.82 | 0.71 | 1.16 | Medium-separated |
| C3 | C6 | 0.82 | 0.7 | 1.18 | Medium-separated |
| C3 | C7 | 0.84 | 0.54 | 1.56 | Far-separated |
| C3 | C8 | 0.83 | 0.72 | 1.15 | Medium-separated |
| C3 | C9 | 0.83 | 0.64 | 1.3 | Well-separated |
| C3 | C10 | 0.83 | 0.67 | 1.25 | Well-separated |
| C3 | C11 | 0.84 | 0.62 | 1.35 | Well-separated |
| C3 | C12 | 0.81 | 0.7 | 1.16 | Medium-separated |
| C3 | C13 | 0.84 | 0.63 | 1.32 | Well-separated |
| C3 | C14 | 0.82 | 0.69 | 1.19 | Medium-separated |
| C3 | C15 | 0.83 | 0.69 | 1.21 | Medium-separated |
| C3 | C16 | 0.8 | 0.69 | 1.16 | Medium-separated |
| C3 | C17 | 0.83 | 0.64 | 1.28 | Well-separated |
| C3 | C18 | 0.82 | 0.69 | 1.19 | Medium-separated |
| C3 | C19 | 0.82 | 0.7 | 1.17 | Medium-separated |
| C3 | C20 | 0.82 | 0.69 | 1.18 | Medium-separated |
| C3 | C21 | 0.81 | 0.7 | 1.16 | Medium-separated |
| C3 | C22 | 0.82 | 0.7 | 1.17 | Medium-separated |
| C3 | C23 | 0.83 | 0.68 | 1.22 | Medium-separated |
| C3 | C24 | 0.81 | 0.7 | 1.16 | Medium-separated |
| C3 | C25 | 0.82 | 0.7 | 1.17 | Medium-separated |
| C3 | C26 | 0.81 | 0.69 | 1.17 | Medium-separated |
| C4 | C5 | 0.82 | 0.71 | 1.15 | Medium-separated |
| C4 | C6 | 0.81 | 0.69 | 1.17 | Medium-separated |
| C4 | C7 | 0.83 | 0.53 | 1.55 | Far-separated |
| C4 | C8 | 0.82 | 0.71 | 1.15 | Medium-separated |
| C4 | C9 | 0.85 | 0.64 | 1.33 | Well-separated |
| C4 | C10 | 0.83 | 0.66 | 1.26 | Well-separated |

| C4 | C11 | 0.82 | 0.62 | 1.34 | Well-separated |
|----|-----|------|------|------|----------------|
| C4 | C12 | 0.8 | 0.69 | 1.16 | Medium-separated |
| C4 | C13 | 0.82 | 0.63 | 1.31 | Well-separated |
| C4 | C14 | 0.81 | 0.69 | 1.18 | Medium-separated |
| C4 | C15 | 0.82 | 0.68 | 1.2 | Medium-separated |
| C4 | C16 | 0.81 | 0.7 | 1.16 | Medium-separated |
| C4 | C17 | 0.82 | 0.64 | 1.28 | Well-separated |
| C4 | C18 | 0.82 | 0.69 | 1.18 | Medium-separated |
| C4 | C19 | 0.81 | 0.7 | 1.16 | Medium-separated |
| C4 | C20 | 0.81 | 0.69 | 1.17 | Medium-separated |
| C4 | C21 | 0.8 | 0.7 | 1.16 | Medium-separated |
| C4 | C22 | 0.82 | 0.7 | 1.16 | Medium-separated |
| C4 | C23 | 0.82 | 0.68 | 1.21 | Medium-separated |
| C4 | C24 | 0.81 | 0.7 | 1.15 | Medium-separated |
| C4 | C25 | 0.82 | 0.7 | 1.16 | Medium-separated |
| C4 | C26 | 0.81 | 0.69 | 1.17 | Medium-separated |
| C5 | C6 | 0.81 | 0.79 | 1.03 | Less-separated |
| C5 | C7 | 0.81 | 0.6 | 1.34 | Well-separated |
| C5 | C8 | 0.83 | 0.81 | 1.02 | Less-separated |
| C5 | C10 | 0.82 | 0.72 | 1.14 | Medium-separated |
| C5 | C11 | 0.79 | 0.67 | 1.17 | Medium-separated |
| C5 | C12 | 0.82 | 0.71 | 1.16 | Medium-separated |
| C5 | C13 | 0.78 | 0.7 | 1.12 | Medium-separated |
| C5 | C14 | 0.82 | 0.77 | 1.05 | Less-separated |
| C5 | C15 | 0.82 | 0.71 | 1.14 | Medium-separated |
| C5 | C16 | 0.82 | 0.71 | 1.15 | Medium-separated |
| C5 | C17 | 0.88 | 0.73 | 1.21 | Medium-separated |
| C5 | C18 | 0.81 | 0.72 | 1.12 | Medium-separated |
| C5 | C19 | 0.79 | 0.76 | 1.04 | Less-separated |
| C5 | C21 | 0.82 | 0.71 | 1.16 | Medium-separated |
| C5 | C23 | 0.84 | 0.8 | 1.06 | Less-separated |
| C5 | C24 | 0.83 | 0.73 | 1.14 | Medium-separated |
| C5 | C25 | 0.82 | 0.79 | 1.04 | Less-separated |
| C5 | C26 | 0.8 | 0.72 | 1.12 | Medium-separated |
| C6 | C7 | 0.78 | 0.58 | 1.36 | Well-separated |
| C6 | C8 | 0.82 | 0.79 | 1.03 | Less-separated |
| C6 | C10 | 0.81 | 0.7 | 1.17 | Medium-separated |
| C6 | C11 | 0.75 | 0.63 | 1.19 | Medium-separated |
| C6 | C12 | 0.82 | 0.69 | 1.18 | Medium-separated |
| C6 | C13 | 0.73 | 0.64 | 1.14 | Medium-separated |
| C6 | C15 | 0.79 | 0.67 | 1.17 | Medium-separated |
| C6 | C16 | 0.81 | 0.69 | 1.18 | Medium-separated |
| C6 | C18 | 0.77 | 0.67 | 1.15 | Medium-separated |
| C6 | C19 | 0.77 | 0.72 | 1.06 | Less-separated |
| C6 | C21 | 0.82 | 0.69 | 1.18 | Medium-separated |
| C6 | C24 | 0.82 | 0.7 | 1.16 | Medium-separated |
| C6 | C26 | 0.8 | 0.69 | 1.15 | Medium-separated |
| C7 | C8 | 0.81 | 0.6 | 1.34 | Well-separated |
| C7 | C9 | 0.81 | 0.61 | 1.33 | Well-separated |
| C7 | C10 | 0.84 | 0.6 | 1.41 | Far-separated |
| C7 | C11 | 0.81 | 0.6 | 1.35 | Well-separated |
| C7 | C12 | 0.84 | 0.54 | 1.56 | Far-separated |
| C7 | C13 | 0.81 | 0.61 | 1.33 | Well-separated |
| C7 | C14 | 0.79 | 0.57 | 1.38 | Well-separated |
| C7 | C15 | 0.81 | 0.53 | 1.54 | Far-separated |
| C7 | C16 | 0.83 | 0.54 | 1.56 | Far-separated |
| C7 | C17 | 0.83 | 0.58 | 1.43 | Far-separated |
| C7 | C18 | 0.8 | 0.53 | 1.51 | Far-separated |
| C7 | C19 | 0.82 | 0.58 | 1.4 | Well-separated |
| C7 | C20 | 0.8 | 0.58 | 1.38 | Well-separated |
| C7 | C21 | 0.84 | 0.54 | 1.56 | Far-separated |
| C7 | C22 | 0.81 | 0.6 | 1.36 | Well-separated |
| C7 | C23 | 0.82 | 0.59 | 1.4 | Well-separated |
| C7 | C24 | 0.83 | 0.54 | 1.53 | Far-separated |
| C7 | C25 | 0.8 | 0.59 | 1.36 | Well-separated |
| C7 | C26 | 0.84 | 0.55 | 1.54 | Far-separated |
| C8 | C9 | 0.92 | 0.81 | 1.14 | Medium-separated |
| C8 | C10 | 0.83 | 0.72 | 1.14 | Medium-separated |
| C8 | C11 | 0.79 | 0.68 | 1.17 | Medium-separated |
| C8 | C12 | 0.82 | 0.71 | 1.15 | Medium-separated |
| C8 | C13 | 0.78 | 0.7 | 1.12 | Medium-separated |
| C8 | C14 | 0.82 | 0.78 | 1.05 | Less-separated |
| C8 | C15 | 0.82 | 0.72 | 1.14 | Medium-separated |
| C8 | C16 | 0.82 | 0.71 | 1.15 | Medium-separated |
| C8 | C17 | 0.87 | 0.73 | 1.19 | Medium-separated |
| C8 | C18 | 0.81 | 0.72 | 1.12 | Medium-separated |
| C8 | C19 | 0.79 | 0.76 | 1.04 | Less-separated |
| C8 | C20 | 0.83 | 0.8 | 1.03 | Less-separated |
| C8 | C21 | 0.82 | 0.71 | 1.15 | Medium-separated |
| C8 | C22 | 0.83 | 0.81 | 1.02 | Less-separated |
| C8 | C23 | 0.84 | 0.8 | 1.05 | Less-separated |
| C8 | C24 | 0.83 | 0.73 | 1.13 | Medium-separated |
| C8 | C25 | 0.82 | 0.79 | 1.04 | Less-separated |
| C8 | C26 | 0.81 | 0.72 | 1.12 | Medium-separated |
| C9 | C10 | 0.88 | 0.6 | 1.46 | Far-separated |
| C9 | C11 | 0.86 | 0.58 | 1.48 | Far-separated |

| | | | | | |
|---|---|---|---|---|---|
| C9 | C12 | 0.83 | 0.64 | 1.3 | Well-separated |
| C9 | C13 | 0.88 | 0.6 | 1.48 | Far-separated |
| C9 | C15 | 0.97 | 0.68 | 1.44 | Far-separated |
| C9 | C16 | 0.84 | 0.64 | 1.31 | Well-separated |
| C9 | C18 | 0.98 | 0.71 | 1.38 | Well-separated |
| C9 | C19 | 0.99 | 0.69 | 1.44 | Far-separated |
| C9 | C21 | 0.83 | 0.64 | 1.3 | Well-separated |
| C9 | C24 | 0.9 | 0.66 | 1.36 | Well-separated |
| C9 | C26 | 0.86 | 0.63 | 1.37 | Well-separated |
| C10 | C11 | 0.83 | 0.68 | 1.22 | Medium-separated |
| C10 | C12 | 0.83 | 0.66 | 1.25 | Well-separated |
| C10 | C13 | 0.83 | 0.69 | 1.21 | Medium-separated |
| C10 | C14 | 0.82 | 0.7 | 1.17 | Medium-separated |
| C10 | C15 | 0.82 | 0.65 | 1.25 | Well-separated |
| C10 | C16 | 0.83 | 0.66 | 1.25 | Well-separated |
| C10 | C17 | 0.84 | 0.68 | 1.23 | Medium-separated |
| C10 | C18 | 0.81 | 0.66 | 1.23 | Medium-separated |
| C10 | C19 | 0.84 | 0.69 | 1.2 | Medium-separated |
| C10 | C20 | 0.83 | 0.7 | 1.18 | Medium-separated |
| C10 | C21 | 0.83 | 0.66 | 1.25 | Well-separated |
| C10 | C22 | 0.83 | 0.71 | 1.16 | Medium-separated |
| C10 | C23 | 0.84 | 0.68 | 1.24 | Medium-separated |
| C10 | C24 | 0.83 | 0.67 | 1.24 | Medium-separated |
| C10 | C25 | 0.82 | 0.7 | 1.18 | Medium-separated |
| C10 | C26 | 0.84 | 0.67 | 1.26 | Well-separated |
| C11 | C12 | 0.83 | 0.62 | 1.35 | Well-separated |
| C11 | C13 | 0.75 | 0.63 | 1.2 | Medium-separated |
| C11 | C14 | 0.75 | 0.62 | 1.2 | Medium-separated |
| C11 | C15 | 0.8 | 0.6 | 1.33 | Well-separated |
| C11 | C16 | 0.83 | 0.62 | 1.34 | Well-separated |
| C11 | C17 | 0.78 | 0.63 | 1.24 | Medium-separated |
| C11 | C18 | 0.78 | 0.6 | 1.32 | Well-separated |
| C11 | C19 | 0.79 | 0.65 | 1.2 | Medium-separated |
| C11 | C20 | 0.76 | 0.63 | 1.21 | Medium-separated |
| C11 | C21 | 0.83 | 0.62 | 1.35 | Well-separated |
| C11 | C22 | 0.77 | 0.65 | 1.19 | Medium-separated |
| C11 | C23 | 0.81 | 0.64 | 1.26 | Well-separated |
| C11 | C24 | 0.82 | 0.62 | 1.31 | Well-separated |
| C11 | C25 | 0.78 | 0.64 | 1.2 | Medium-separated |
| C11 | C26 | 0.83 | 0.63 | 1.31 | Well-separated |
| C12 | C13 | 0.83 | 0.63 | 1.32 | Well-separated |
| C12 | C14 | 0.82 | 0.69 | 1.18 | Medium-separated |
| C12 | C15 | 0.83 | 0.69 | 1.2 | Medium-separated |
| C12 | C16 | 0.8 | 0.69 | 1.16 | Medium-separated |
| C12 | C17 | 0.82 | 0.64 | 1.28 | Well-separated |
| C12 | C18 | 0.82 | 0.69 | 1.19 | Medium-separated |
| C12 | C19 | 0.81 | 0.7 | 1.17 | Medium-separated |
| C12 | C20 | 0.81 | 0.69 | 1.18 | Medium-separated |
| C12 | C21 | 0.81 | 0.7 | 1.16 | Medium-separated |
| C12 | C22 | 0.82 | 0.7 | 1.17 | Medium-separated |
| C12 | C23 | 0.83 | 0.68 | 1.22 | Medium-separated |
| C12 | C24 | 0.81 | 0.7 | 1.15 | Medium-separated |
| C12 | C25 | 0.82 | 0.7 | 1.17 | Medium-separated |
| C12 | C26 | 0.81 | 0.69 | 1.17 | Medium-separated |
| C13 | C14 | 0.74 | 0.64 | 1.15 | Medium-separated |
| C13 | C15 | 0.79 | 0.61 | 1.3 | Well-separated |
| C13 | C16 | 0.83 | 0.63 | 1.32 | Well-separated |
| C13 | C17 | 0.78 | 0.64 | 1.21 | Medium-separated |
| C13 | C18 | 0.78 | 0.61 | 1.28 | Well-separated |
| C13 | C19 | 0.78 | 0.66 | 1.18 | Medium-separated |
| C13 | C20 | 0.74 | 0.65 | 1.14 | Medium-separated |
| C13 | C21 | 0.83 | 0.63 | 1.32 | Well-separated |
| C13 | C22 | 0.76 | 0.67 | 1.13 | Medium-separated |
| C13 | C23 | 0.8 | 0.65 | 1.23 | Medium-separated |
| C13 | C24 | 0.82 | 0.64 | 1.28 | Well-separated |
| C13 | C25 | 0.77 | 0.67 | 1.15 | Medium-separated |
| C13 | C26 | 0.83 | 0.64 | 1.29 | Well-separated |
| C14 | C15 | 0.8 | 0.7 | 1.15 | Medium-separated |
| C14 | C16 | 0.82 | 0.69 | 1.18 | Medium-separated |
| C14 | C18 | 0.79 | 0.69 | 1.14 | Medium-separated |
| C14 | C19 | 0.78 | 0.72 | 1.07 | Less-separated |
| C14 | C21 | 0.82 | 0.69 | 1.19 | Medium-separated |
| C14 | C24 | 0.82 | 0.7 | 1.17 | Medium-separated |
| C14 | C26 | 0.8 | 0.69 | 1.15 | Medium-separated |
| C15 | C16 | 0.82 | 0.68 | 1.21 | Medium-separated |
| C15 | C17 | 0.82 | 0.61 | 1.35 | Well-separated |
| C15 | C18 | 0.8 | 0.7 | 1.15 | Medium-separated |
| C15 | C19 | 0.8 | 0.7 | 1.14 | Medium-separated |
| C15 | C20 | 0.81 | 0.7 | 1.15 | Medium-separated |
| C15 | C21 | 0.83 | 0.68 | 1.21 | Medium-separated |
| C15 | C22 | 0.81 | 0.7 | 1.16 | Medium-separated |
| C15 | C23 | 0.82 | 0.66 | 1.24 | Medium-separated |
| C15 | C24 | 0.83 | 0.7 | 1.19 | Medium-separated |
| C15 | C25 | 0.82 | 0.72 | 1.14 | Medium-separated |
| C15 | C26 | 0.82 | 0.7 | 1.18 | Medium-separated |
| C16 | C17 | 0.82 | 0.64 | 1.28 | Well-separated |

| | | | | | |
|---|---|---|---|---|---|
| C16 | C18 | 0.82 | 0.69 | 1.19 | Medium-separated |
| C16 | C19 | 0.82 | 0.7 | 1.17 | Medium-separated |
| C16 | C20 | 0.81 | 0.69 | 1.18 | Medium-separated |
| C16 | C21 | 0.81 | 0.7 | 1.16 | Medium-separated |
| C16 | C22 | 0.82 | 0.7 | 1.16 | Medium-separated |
| C16 | C23 | 0.82 | 0.68 | 1.22 | Medium-separated |
| C16 | C24 | 0.81 | 0.7 | 1.15 | Medium-separated |
| C16 | C25 | 0.82 | 0.7 | 1.17 | Medium-separated |
| C16 | C26 | 0.81 | 0.69 | 1.17 | Medium-separated |
| C17 | C18 | 0.81 | 0.6 | 1.34 | Well-separated |
| C17 | C19 | 0.81 | 0.7 | 1.16 | Medium-separated |
| C17 | C21 | 0.82 | 0.64 | 1.28 | Well-separated |
| C17 | C24 | 0.83 | 0.66 | 1.27 | Well-separated |
| C17 | C26 | 0.81 | 0.66 | 1.24 | Medium-separated |
| C18 | C19 | 0.79 | 0.71 | 1.11 | Medium-separated |
| C18 | C20 | 0.79 | 0.71 | 1.12 | Medium-separated |
| C18 | C21 | 0.82 | 0.69 | 1.19 | Medium-separated |
| C18 | C22 | 0.8 | 0.71 | 1.13 | Medium-separated |
| C18 | C23 | 0.81 | 0.67 | 1.21 | Medium-separated |
| C18 | C24 | 0.82 | 0.7 | 1.17 | Medium-separated |
| C18 | C25 | 0.83 | 0.74 | 1.12 | Medium-separated |
| C18 | C26 | 0.81 | 0.7 | 1.15 | Medium-separated |
| C19 | C20 | 0.78 | 0.73 | 1.06 | Less-separated |
| C19 | C21 | 0.82 | 0.7 | 1.17 | Medium-separated |
| C19 | C22 | 0.8 | 0.76 | 1.05 | Less-separated |
| C19 | C23 | 0.78 | 0.73 | 1.07 | Less-separated |
| C19 | C24 | 0.82 | 0.71 | 1.15 | Medium-separated |
| C19 | C25 | 0.78 | 0.72 | 1.08 | Less-separated |
| C19 | C26 | 0.8 | 0.7 | 1.13 | Medium-separated |
| C20 | C21 | 0.81 | 0.69 | 1.18 | Medium-separated |
| C20 | C24 | 0.82 | 0.7 | 1.16 | Medium-separated |
| C20 | C26 | 0.79 | 0.7 | 1.14 | Medium-separated |
| C21 | C22 | 0.82 | 0.7 | 1.17 | Medium-separated |
| C21 | C23 | 0.83 | 0.68 | 1.22 | Medium-separated |
| C21 | C24 | 0.81 | 0.7 | 1.15 | Medium-separated |
| C21 | C25 | 0.82 | 0.7 | 1.17 | Medium-separated |
| C21 | C26 | 0.81 | 0.69 | 1.17 | Medium-separated |
| C22 | C24 | 0.83 | 0.72 | 1.15 | Medium-separated |
| C22 | C26 | 0.8 | 0.71 | 1.14 | Medium-separated |
| C23 | C24 | 0.83 | 0.7 | 1.19 | Medium-separated |
| C23 | C26 | 0.8 | 0.68 | 1.18 | Medium-separated |
| C24 | C25 | 0.83 | 0.72 | 1.15 | Medium-separated |
| C24 | C26 | 0.82 | 0.71 | 1.16 | Medium-separated |
| C25 | C26 | 0.8 | 0.71 | 1.13 | Medium-separated |
| C2 | - | 0.73 | 0.64 | 1.14 | - |
| C5 | - | 0.82 | 0.8 | 1.02 | - |
| C6 | - | 0.85 | 0.65 | 1.32 | - |
| C14 | - | 0.78 | 0.71 | 1.11 | - |
| C17 | - | | 0.55 | | - |
| C22 | - | 0.87 | 0.84 | 1.04 | - |
| C23 | - | | 0.53 | | - |
| C25 | - | 0.82 | 0.78 | 1.05 | - |
| C2 | C9 | 0.96 | 0.68 | 1.42 | - |
| C2 | C20 | 0.76 | 0.67 | 1.13 | - |
| C5 | C9 | 0.94 | 0.82 | 1.14 | - |
| C5 | C20 | 0.83 | 0.81 | 1.03 | - |
| C5 | C22 | 0.82 | 0.81 | 1.02 | - |
| C6 | C9 | 1 | 0.77 | 1.29 | - |
| C6 | C14 | 0.78 | 0.69 | 1.13 | - |
| C6 | C17 | 1.15 | 0.67 | 1.71 | - |
| C6 | C20 | 0.83 | 0.74 | 1.12 | - |
| C6 | C22 | 0.92 | 0.83 | 1.12 | - |
| C6 | C23 | 0.87 | 0.76 | 1.15 | - |
| C6 | C25 | 0.76 | 0.71 | 1.08 | - |
| C9 | C14 | 0.82 | 0.51 | 1.6 | - |
| C9 | C17 | 0.98 | 0.56 | 1.75 | - |
| C9 | C20 | 0.63 | 0.6 | 1.04 | - |
| C9 | C22 | 1.06 | 0.93 | 1.14 | - |
| C9 | C23 | 0.73 | 0.57 | 1.28 | - |
| C9 | C25 | 1.08 | 0.87 | 1.24 | - |
| C14 | C17 | 0.99 | 0.6 | 1.64 | - |
| C14 | C20 | 0.87 | 0.77 | 1.13 | - |
| C14 | C22 | 0.79 | 0.71 | 1.11 | - |
| C14 | C23 | 0.83 | 0.67 | 1.23 | - |
| C14 | C25 | 0.75 | 0.7 | 1.07 | - |
| C17 | C20 | 0.55 | 0.53 | 1.04 | - |
| C17 | C22 | 1.23 | 0.7 | 1.75 | - |
| C17 | C23 | 1.39 | 0.59 | 2.35 | - |
| C17 | C25 | 0.88 | 0.64 | 1.38 | - |
| C20 | C22 | 0.88 | 0.85 | 1.04 | - |
| C20 | C23 | 0.86 | 0.74 | 1.16 | - |
| C20 | C25 | 0.82 | 0.77 | 1.06 | - |
| C22 | C23 | 0.85 | 0.77 | 1.11 | - |
| C22 | C25 | 0.8 | 0.75 | 1.06 | - |
| C23 | C25 | 0.81 | 0.73 | 1.11 | - |

