# OpenReview forum: "A Benchmark Dataset for Learning from Label Proportions"
_ICLR.cc/2023/Conference — Submitted to ICLR 2023_

### Official Review · Reviewer_w1Wh · 2022-10-20

**Confidence:** 3
**Correctness:** 4
**Technical Novelty And Significance:** 3
**Empirical Novelty And Significance:** Not applicable
**Recommendation:** 5

**Clarity, Quality, Novelty And Reproducibility:**

Overall, this submission provides the enough details of the generated LLP artificial benchmark. However, the presentation quality can be greatly improved. In fact, there are many typos and grammatical problems throughout the paper.

**Details Of Ethics Concerns:**

N/A.

**Strength And Weaknesses:**

Strength:
1. According to Table 10 in Appendix A.8, most of the generated LLP benchmarks contains many bags (more than one million for some of them) and their characteristics are diversified.
2. The details of the generation procedure are provided which is convenient for readers.
3. Some analysis and experiments on these generated LLP benchmarks are also provided.

Weaknesses:
The task of learning from label proportions (LLP) is interesting. However, as also stated in this submission: " they typically evaluated their methods on pseudo-synthetically generated LLP training data using common small scale supervised learning datasets by randomly sampling or partitioning their instances into bags.".  This submission claims that a large scale LLP benchmark constructed from the Criteo Kaggle CTR dataset is proposed. Maybe the benchmark is really large, but it is also artificially generated from binary classification data set (i.e., Criteo Kaggle CTR dataset), not a real-world LLP data set. I don't know whether this will make sense to generate LLP artificial benchmarks.

**Summary Of The Paper:**

In this paper, authors generated some artificial benchmarks for the tasks of learning from label proportions (LLP) based on the publicly-available Criteo Kaggle CTR dataset.

**Summary Of The Review:**

As I have stated before, the main contribution of this submission is to generating some artificial benchmarks for the tasks of learning from label proportions (LLP) based on the publicly-available Criteo Kaggle CTR dataset. Maybe the generated benchmarks are large, but they are still artificial data sets (though generated from real-world binary classification data set). I am not sure that such contribution is deserved to be accepted by ICLR. I'd love to hear from other reviewers.

---

### Official Review · Reviewer_AGL5 · 2022-10-25

**Confidence:** 3
**Correctness:** 3
**Technical Novelty And Significance:** 1
**Empirical Novelty And Significance:** 2
**Recommendation:** 3

**Clarity, Quality, Novelty And Reproducibility:**

The submission is quite clear and easy to read. All the information appears to be available.


**Strength And Weaknesses:**

The submission purports to introduce a natural large-scale dataset for learning from label proportions. However, given the absence of any information involving the categorical features used to establish bags, regardless of the summary statistics produced, it seems entirely unclear how natural or realistic the constructed data is.

The number of LLP methods considered in the experiments seems quite limited.

Other comments and questions:

"large number of mostly artificial methods of forming bags from instance level datasets." - why is this a drawback?

Please define what CTR means.

"to the partition the groupings "

Is the squared Euclidean distance calculated based on the numeric features only (for inter- and intra-bag distances)?

What is the rationale for performing the 5-fold cross-validation at the instance-level before re-creating bags?

"techniques such neural"

"The model predicts on all the instances in the bags of the minibatch are aggregated"

" over the for the"

Why not use a distance metric for bags that has been used in the past in machine learning, based on the Hausdorff distance?

"The categorical features are encoded as non-negative integers." - this implies an order

" it’s embedding layer "

"which we transform the instances into this embedding space"

The results for SGD do not seem to add anything to the paper.


**Summary Of The Paper:**

The submission presents a collection of benchmark datasets for learning from label proportions. It carefully describes how all these datasets were constructed from the original data in the Criteo dataset, by computing bags of instances based on groupings induced by categorical variables in the data. The submission presents a substantial number summary statistics for the constructed benchmark datasets. It also compares a small number of learning algorithms on this data.


**Summary Of The Review:**

Using a dataset with unknown semantics of the features to create an LLP problem by grouping instances according to these features does not yield a compelling benchmark. Also, only a small number of LLP methods are compared on the benchmark.

---

### Official Review · Reviewer_A9So · 2022-11-01

**Confidence:** 3
**Clarity, Quality, Novelty And Reproducibility:** The presentation of the paper is clea…
**Correctness:** 2
**Technical Novelty And Significance:** 2
**Empirical Novelty And Significance:** 2
**Recommendation:** 3

**Strength And Weaknesses:**

Strength:

- A detailed analysis of the differences among grouping criterion


Weakness:

- The paper is incremental from Saket et al. (2022). Basically, it computes some statistics on the different groups with the same dataset in previous work.
- A key question on how to balance the bag size and performance is not answered. The paper only reports that smaller bag size generally leads to better performance, but didn't quantify how larger bag size harms privacy on the benchmark dataset. Therefore, it's hard to be motivated to choose larger bag sizes.
- The numbers/ thresholds used in 5.1 seems to be picked without ground.
- The dataset is too toy. The method performed on this 39-dimensional dataset may not be able to generalize on image/text data.
- Is there any proof showing that the method performing better on this dataset can be generalized to other datasets?

**Summary Of The Paper:**

The paper created a benchmark dataset for LLP problem, where people would like to train an instance predictor based on bags of features and positive ratios in the bag. The main contribution is to split bags with feature sets, analyze the statistics of bags, and benchmark the performances of different LLP methods.

**Summary Of The Review:**

In summary, it's not well-motivated to construct such a dataset. The proposed dataset didn't provide any instruction on what kind of model will be better for LLP.

---

### Official Review · Reviewer_fQUW · 2022-11-01

**Confidence:** 4
**Correctness:** 3
**Technical Novelty And Significance:** 2
**Empirical Novelty And Significance:** 2
**Recommendation:** 3

**Clarity, Quality, Novelty And Reproducibility:**

The presentation of this paper is not so clear and needs improvement.
The purpose of this paper is to build a large-scale benchmark for LLP, which is the innovation of this paper. But the reason for using the Criteo Kaggle CTR dataset and grouping in this way needs a clearer explanation. At the same time, directly using the existing category labels of the Criteo Kaggle CTR dataset for generating bags is insufficiently innovative. Whether this dataset can contribute to the field of LLP requires more experimental verification.


**Strength And Weaknesses:**

Pros:
- The authors considered establishing a benchmark dataset for LLP, which did not exist in previous studies.

Cons:
 - It is a bit confusing why to choose the Criteo Kaggle CTR dataset and why the dataset groups in this way. First of all, in the Criteo Kaggle CTR dataset, the semantics of these 26 category features are undisclosed and missing for some samples, so the direct use of category features for subcontracting requires a more reasonable explanation. Secondly, why using at most two categorical features for grouping also needs a reasonable explanation.
In LLP, the natural grouping of bags can usually be based on geographic location, time point, age group, etc. It may be more appropriate to select a dataset with these kinds of information.
- In the study of LLP, the performance of various techniques on bags of different sizes is also very important. The reason for removing bags with a size greater than 2500 or less than 50 needs to be explained.
- Grouping based on category features for the Criteo Kaggle CTR dataset has existed in the work of Saket et al. (2022). This is slightly insufficient as the innovation point of this paper.
- The authors used the test AUC score to qualify the tractability of the LLP dataset, but the result did not seem to be ideal (AUC scores were centered around 60-80%). It would be nice to add some experiments that use SOTA methods, such as LLP-VAT (Tsai et al. 2020) or MCM (Scott et al. 2020), to illustrate that the dataset makes sense.

**Summary Of The Paper:**

This paper focuses on creating a benchmark as a collection of diverse and large-scale LLP datasets based on the Criteo Kaggle CTR dataset. The authors obtained all bag sets grouped by one or two categorical features and applied two filters on the clipped groupings to choose groupings for model training. They estimated the geometric clustering of bags and qualified the tractability of an LLP dataset by the test AUC scores.

**Summary Of The Review:**

The novelty of this paper is insufficient. It still has room for improvement.

---

### Decision · Program_Chairs · 2023-01-20

**Decision:**

Reject

**Justification For Why Not Higher Score:**

Thanks for the clarification for the reviewers' questions, which resolved some of the issues raised by their initial reviews. Nevertheless, I still second Reviewer A9So and think it is rather unclear how natural or realistic the constructed data is, under the absence of any information involving the categorical features used to establish bags. For this reason, the current manuscript is not suitable for publication.

**Justification For Why Not Lower Score:**

N?A

**Metareview: Summary, Strengths And Weaknesses:**

Summary:
The authors propose benchmark datasets for learning from label proportions, which is an important problem. Their datasets are based on the Criteo Kaggle CTR dataset.

Strength:
Benchmark datasets for learning from label proportions are novel.

Weakness:
Rather incremental from Saket et al. (2022).